# Regulation of positive and negative selection and TCR signaling during thymic T cell development by capicua

**Soeun Kim[1], Guk-Yeol Park[1], Jong Seok Park[1], Jiho Park[1], Hyebeen Hong[1], Yoontae Lee[1,2]***

[1]Department of Life Sciences, Pohang University of Science and Technology (POSTECH), Pohang, Republic of Korea; [2]Institute of Convergence Science, Yonsei University, Seoul, Republic of Korea

**Abstract** Central tolerance is achieved through positive and negative selection of thymocytes mediated by T cell receptor (TCR) signaling strength. Thus, dysregulation of the thymic selection process often leads to autoimmunity. Here, we show that Capicua (CIC), a transcriptional repressor that suppresses autoimmunity, controls the thymic selection process. Loss of CIC prior to T-cell lineage commitment impairs both positive and negative selection of thymocytes. CIC deficiency attenuated TCR signaling in CD4+CD8+ double-positive (DP) cells, as evidenced by a decrease in CD5 and phospho-ERK levels and calcium flux. We identified *Spry4*, *Dusp4*, *Dusp6*, and *Spred1* as CIC target genes that could inhibit TCR signaling in DP cells. Furthermore, impaired positive selection and TCR signaling were partially rescued in *Cic* and *Spry4* double mutant mice. Our findings indicate that CIC is a transcription factor required for thymic T cell development and suggests that CIC acts at multiple stages of T cell development and differentiation to prevent autoimmunity.

*For correspondence:
yoontael@postech.ac.kr

Competing interest: The authors declare that no competing interests exist.

## Editor's evaluation

This paper focuses on the transcriptional regulation of the T cell receptor (TCR) signaling cascade and would be of interest to those studying T cell development and differentiation. The authors employ a conditional deletion of the Capicua (Cic) gene, a transcriptional repressor previously shown to be involved in regulating autoimmunity and follicular helper T (Tfh) cell differentiation, and now show that loss of CIC in hematopoietic cells leads to defects in TCR-β selection as well as in positive and negative selection of developing thymocytes. The overall conclusions are well supported by the findings.

## Introduction

T cells play a crucial role in the adaptive immune system's defense against external invasion. To distinguish between self and non-self, T cells use their T cell receptors (TCRs) to recognize peptide-loaded major histocompatibility complex (MHC) molecules and respond to many types of antigens with an enormous TCR repertoire. T cells with a specific TCR are generated in the thymus through a series of processes, ranging from random rearrangement of TCR gene segments to selection processes that entail apoptosis of inappropriate cells (*Klein et al., 2014*).

T cell development in the thymus is tightly regulated to prevent the generation of nonfunctional or self-reactive T cells. Progenitor cells from bone marrow (BM) become CD4-CD8- double-negative (DN) cells and undergo a process called β-selection, which selects only T cells with a functional TCRβ chain. DN cells that have passed β-selection then become CD4+CD8+ double-positive (DP) cells and

undergo positive selection, which selects T cells that bind to self-peptide ligands loaded onto MHC (self-pMHC) molecules. These CD4$^+$CD8$^+$ DP cells then develop into CD4$^+$CD8$^-$ single-positive (CD4$^+$ SP) or CD4$^-$CD8$^+$ single-positive (CD8$^+$ SP) T cells. Negative selection, also known as clonal deletion, selectively removes T cells that bind with high affinity to self-pMHC during the DP and SP stages. Because the intensity and duration of TCR signaling based on TCR affinity to self-pMHC are the major determinants of selection (*Gascoigne et al., 2016*), defects in the TCR signaling component lead to abnormal T cell development and alteration of the TCR repertoire (*Fu et al., 2010*; *Sakaguchi et al., 2003*). Impairment of thymic selection caused by decreased TCR signaling destroys central tolerance, and consequently induces autoimmunity accompanied by the expansion of the CD44$^{hi}$CD62L$^{lo}$ activated effector/memory T cell population (*Fu et al., 2010*; *Sakaguchi et al., 2003*; *Sommers et al., 2002*).

Capicua (CIC) is an evolutionarily conserved transcriptional repressor that regulates the receptor tyrosine kinase (RTK) signaling pathway in *Drosophila* and mammals (*Jiménez et al., 2000*; *Jiménez et al., 2012*). CIC is expressed in two different isoforms: long (CIC-L) and short (CIC-S), which differ in their amino termini (*Lee, 2020*). CIC recognizes specific octameric DNA sequences (5'-T(G/C) AATG(A/G)(A/G)–3') within its target gene promoter region and represses its expression (*Kawamura-Saito et al., 2006*; *Shin and Hong, 2014*; *Weissmann et al., 2018*). Several genomic and transcriptomic analyses have identified CIC target genes, including *ETV1, ETV4, ETV5, SPRY4, SPRED1, DUSP4*, and DUSP6, in various cell types of cells (*Fryer et al., 2011*; *Weissmann et al., 2018*; *Yang et al., 2017*). Activation of the RTK/RAS/MAPK signaling pathway phosphorylates and inactivates CIC via dissociation from target gene promoters, cytoplasmic translocation, and/or proteasomal degradation (*Jiménez et al., 2012*; *Keenan et al., 2020*; *Lee, 2020*). CIC also mediates the ERK-DUSP6-negative feedback loop to maintain ERK activity within the physiological range in mammals (*Ren et al., 2020*).

We previously reported that murine CIC deficiency spontaneously induced lymphoproliferative autoimmune-like phenotypes (*Park et al., 2017*). Hematopoietic lineage cell-specific (*Vav1-Cre*-mediated knockout) and T-cell-specific (*Cd4-Cre*-mediated knockout) *Cic*-null mice commonly exhibit autoimmune-like symptoms including hyperglobulinemia, increased serum anti-dsDNA antibody levels, and tissue infiltration of immune cells, accompanied by increased frequency of CD44$^{hi}$CD62L$^{lo}$ T and follicular helper T (Tfh) cells in the spleen (*Park et al., 2017*). However, these phenotypes were more severe in *Cic$^{f/f}$;Vav1-Cre* mice than in *Cic$^{f/f}$;Cd4-Cre* mice. Moreover, enlargement of secondary lymphoid organs was observed in *Cic$^{f/f}$;Vav1-Cre* mice but not in *Cic$^{f/f}$;Cd4-Cre* mice (*Park et al., 2020*; *Park et al., 2017*). These results indicate that deletion of *Cic* alleles in hematopoietic stem and progenitor cells leads to more severe peripheral T-cell hyperactivation and autoimmunity than the *Cd4-Cre*-mediated *Cic* deletion in CD4$^+$CD8$^+$ DP thymocytes. We also reported enhanced peripheral T cell hyperactivation in *Cic$^{f/f}$;Vav1-Cre* mice relative to *Cic$^{f/f}$;Cd4-Cre* mice were caused by CIC deficiency in T cells rather than in other types of immune cells (*Park et al., 2020*), suggesting that this abnormality could be caused by impaired control of early thymic T cell development in *Cic$^{f/f}$;Vav1-Cre* mice. It was reported that the frequency of CD4$^-$CD8$^-$CD44$^+$CD25$^-$ DN1 cells was increased in adult stage-specific *Cic*-null mice (*Tan et al., 2018*). Taken together, these studies suggest that CIC plays a crucial role in thymic T cell development.

In this study, we found that CIC regulates thymic T cell development from the CD4$^-$CD8$^-$ DN stage and positive and negative selection of thymocytes, primarily during the CD4$^+$CD8$^+$ DP stage. TCR signaling was significantly attenuated in DP cells of *Cic$^{f/f}$;Vav1-Cre* mice, thereby impairing both positive and negative selections in *Cic$^{f/f}$;Vav1-Cre* mice. We also identified *Spry4, Dusp4, Dusp6,* and *Spred1* as CIC target genes that could potentially contribute to the reduced TCR signaling strength and impaired thymic selection processes in *Cic$^{f/f}$;Vav1-Cre* mice. Our findings demonstrate that CIC is a critical regulator of TCR signaling in DP cells and thymic T cell development.

## Results

### Changes in the frequencies of thymic T cell subsets over development in *Cic$^{f/f}$;Vav1-Cre* mice

To examine the role of CIC in thymic T cell development, we first evaluated the levels of CIC in multiple subsets of developing thymocytes using homozygous FLAG-tagged *Cic* knock-in (*Cic$^{FLAG/FLAG}$*) mice (*Park et al., 2019*). Flow cytometry for FLAG-CIC revealed that CIC levels were relatively high

in CD4⁻CD8⁻ DN and immature CD8⁺ single positive (ISP) cells than in cells at later developmental stages, such as CD4⁺CD8⁺ DP and CD4⁺ or CD8⁺ SP subsets (*Figure 1A*). We then determined the frequency and number of thymic T cell subsets in *Cic^f/f* (WT), *Cic^f/f;Cd4-Cre*, and *Cic^f/f;Vav1-Cre* mice at 7 weeks of age. The total number of thymocytes was comparable among WT, *Cic^f/f;Cd4-Cre*, and *Cic^f/f;Vav1-Cre* mice (*Figure 1B*). However, T cell development during the DN stage was abnormal in *Cic^f/f;Vav1-Cre* mice, as evidenced by an increased frequency of CD44^hiCD25^hi DN2 and CD44^loCD25^hi DN3 subsets at the expense of the CD44^loCD25^lo DN4 subset (*Figure 1C*). As expected, these changes were not detected in *Cic^f/f;Cd4-Cre* mice (*Figure 1C*), because *Cd4-Cre* was not expressed in DN cells (*Lee et al., 2001*). We also found a slight increase in the frequency of total thymic γδT cells, which are derived from DN3 thymocytes (*Ciofani and Zúñiga-Pflücker, 2010*), in *Cic^f/f;Vav1-Cre* mice (*Figure 1—figure supplement 1A*). However, the frequencies of mature and type 1 and 17γδT cells were comparable among WT, *Cic^f/f;Cd4-Cre*, and *Cic^f/f;Vav1-Cre* mice (*Figure 1—figure supplement 1B and C*).

We previously reported that the frequency of DN, DP, and SP cells was comparable between WT and *Cic^f/f;Vav1-Cre* mice at 9 weeks of age (*Park et al., 2017*). However, at 7 weeks of age, *Cic^f/f;Vav1-Cre* mice exhibited a mild block in the formation of SP cells in the thymus, and the frequency of DP cells was significantly increased in *Cic^f/f;Vav1-Cre* mice compared to WT and *Cic^f/f;Cd4-Cre* mice, whereas that of SP cells decreased (*Figure 1D*). This difference was not statistically significant when calculated using the cell numbers (*Figure 1D*). To clarify whether CIC deficiency affected thymic SP cell formation, we performed the same analysis using WT and *Cic^f/f;Vav1-Cre* mice at 1 week of age. The frequency and number of CD4⁺ and CD8⁺ SP cells were significantly decreased in 1-week-old *Cic^f/f;Vav1-Cre* mice (*Figure 1E*), suggesting that CIC regulates SP cell development in the thymus. Consistent with this result, both CD4⁺ and CD8⁺ T cell populations were substantially reduced in the spleens of 1-week-old *Cic^f/f;Vav1-Cre* mice (*Figure 1—figure supplement 2A*). In accordance with the observations made at 7 weeks of age (*Figure 1C*), a partial block in the DN3-to-DN4 transition was also found in *Cic^f/f;Vav1-Cre* mice at 1 week of age (*Figure 1—figure supplement 2B*). Total SP thymocytes include peripheral T cells that are recirculated into the thymus as well as SP cells matured from DP thymocytes. The proportion of recirculating CD24^loCD73⁺ CD4⁺ SP cells (*Owen et al., 2019*) in the thymus slightly increased in *Cic^f/f;Vav1-Cre* mice that were 1 week old (*Figure 1—figure supplement 3A*), and was similar between WT and *Cic^f/f;Vav1-Cre* mice at 9 weeks old (*Figure 1—figure supplement 3B*). These results suggest that the decreased frequency of CD4⁺ SP thymocytes in 1-week-old *Cic^f/f;Vav1-Cre* mice was not due to reduced accumulation of recirculating CD4⁺ SP cells in the thymus, and that the disappearance of the decrease in the frequency of SP thymocytes in *Cic^f/f;Vav1-Cre* mice at 9 weeks old (*Park et al., 2017*) might not have resulted from the differential accumulation of recirculating peripheral T cells in the thymus between WT and *Cic^f/f;Vav1-Cre* mice. Together, these data demonstrate that CIC is involved in the regulation of thymic T cell development.

## Stable CIC expression in DP cells of *Cic^f/f;Cd4-Cre* mice

After obtaining the results shown in *Figure 1D*, we wondered why the frequency of DP cells was comparable between WT and *Cic^f/f;Cd4-Cre* mice because *Cic* alleles were supposed to be deleted in DP cells by *Cd4-Cre* (*Lee et al., 2001*). Western blotting for CIC in multiple developing thymic T cell subsets from WT, *Cic^f/f;Cd4-Cre*, and *Cic^f/f;Vav1-Cre* mice provided the answer. Although CIC expression disappeared in all tested thymic T cell subsets of *Cic^f/f;Vav1-Cre* mice, CIC was still substantially expressed in the DP cells of *Cic^f/f;Cd4-Cre* mice (*Figure 2A*). This appeared to be caused by CIC protein stability, because the *loxP* site-flanked genomic regions containing exons 9–11 of *Cic* were completely removed in DP cells by *Cd4-Cre* and *Vav1-Cre* (*Figure 2B*). These data suggest that the impaired SP cell formation in the thymus of *Cic^f/f;Vav1-Cre* mice is attributed to the loss of CIC in DN and/or DP cells rather than that in SP cells, because CIC expression was not detected in SP thymocytes from *Cic^f/f;Cd4-Cre* and *Cic^f/f;Vav1-Cre* mice (*Figure 2A*).

## Impaired thymic positive selection in *Cic^f/f;Vav1-Cre* mice

Defects in thymic positive selection that occurs during the CD4⁺CD8⁺ DP developmental stage often result in a decrease in the CD4⁺ and CD8⁺ SP cell populations (*Fischer et al., 2005*; *Lesourne et al., 2009*; *Neilson et al., 2004*; *Wang et al., 2012*). Therefore, we examined thymic positive selection in *Cic^f/f;Vav1-Cre* mice. First, we analyzed thymocytes from 7-week-old WT, *Cic^f/f;Cd4-Cre*, and

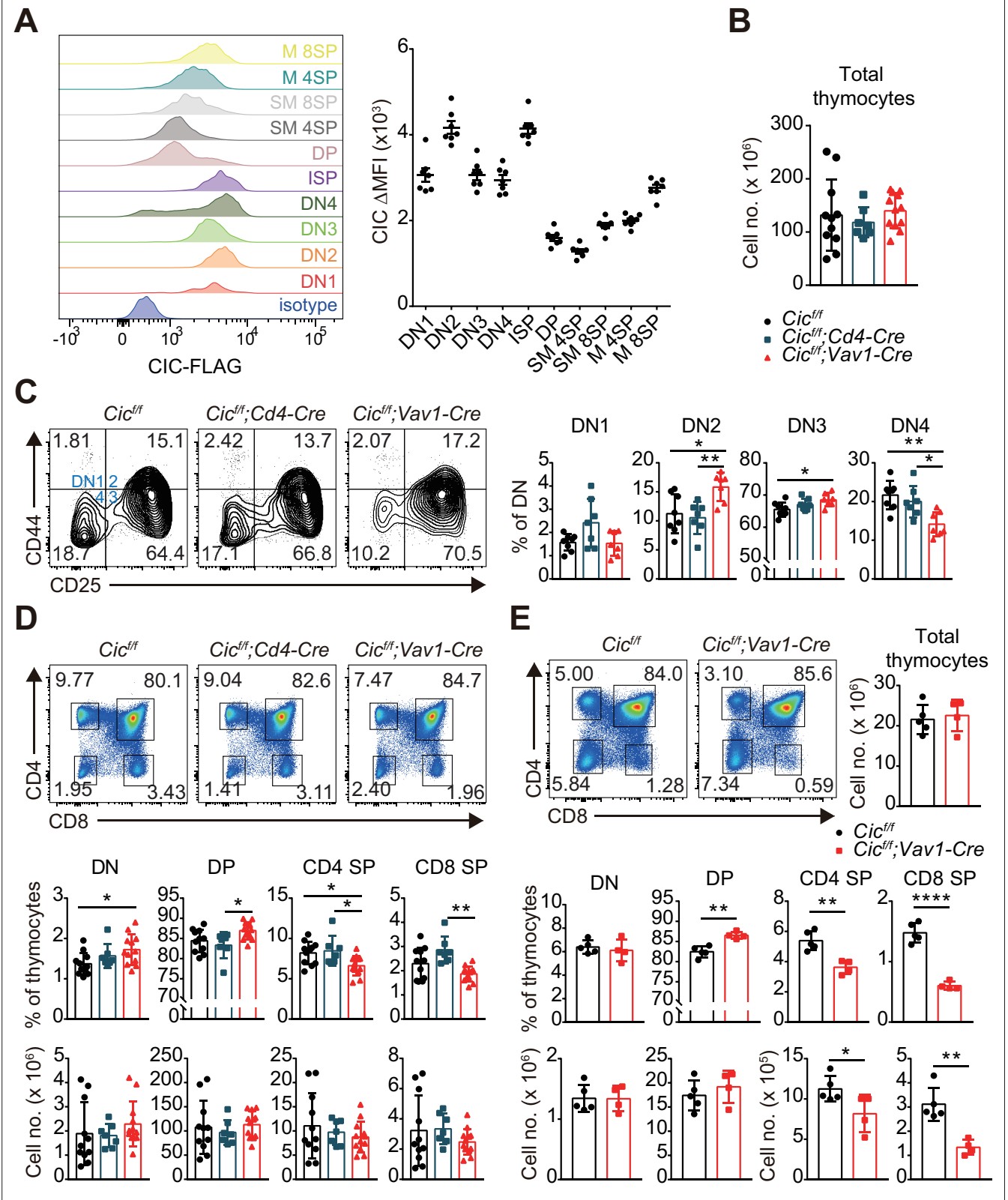

**Figure 1.** Altered T cell development in *Cic^f/f^;Vav1-Cre* mice. (**A**) Capicua (CIC) protein levels in thymic T cell subsets. Thymocytes of *Cic^FLAG/FLAG^* mice (N = 7) were subjected to flow cytometry using anti-FLAG antibody. Representative histograms of CIC-FLAG expression are shown in the left panel for each cell population. The difference in mean fluorescence intensity (ΔMFI) of the CIC-FLAG signal was calculated by subtraction of the MFI value of the isotype control from that obtained by anti-FLAG antibody staining. DN1: CD4⁻CD8⁻CD44^hi^CD25^lo^, DN2: CD4⁻CD8⁻CD44^hi^CD25^hi^, DN3: CD4⁻

*Figure 1 continued on next page*

*Figure 1 continued*

CD8$^-$CD44$^{lo}$CD25$^{hi}$, DN4: CD4$^-$CD8$^-$CD44$^{lo}$CD25$^{lo}$, ISP: immature CD8$^+$ single positive cells (CD4$^-$CD8$^+$TCRβ$^{lo}$CD24$^{hi}$), DP: CD4$^+$CD8$^+$, SM: semi-mature (CD69$^+$TCRβ$^{hi}$), and M: mature (CD69$^-$TCRβ$^{hi}$). Lineage (CD11b, CD11c, CD19, NK1.1, Gr-1, γδTCR, and TER119)-negative (Lin$^-$)-gated cells were analyzed for DN cell populations. (**B–D**) Flow cytometric analysis of thymocytes from 7-week-old *Cic$^{f/f}$*, *Cic$^{f/f}$;Cd4-Cre*, and *Cic$^{f/f}$;Vav1-Cre* mice. (**B**) Total numbers of thymocytes for each genotype. N = 11, 7, and 12 for *Cic$^{f/f}$*, *Cic$^{f/f}$;Cd4-Cre*, and *Cic$^{f/f}$;Vav1-Cre* mice, respectively. (**C**) Proportions of DN1-4 subsets for each genotype. N = 8, 7, and seven for *Cic$^{f/f}$*, *Cic$^{f/f}$;Cd4-Cre*, and *Cic$^{f/f}$;Vav1-Cre* mice, respectively. Lin$^-$-gated cells were used for analysis of DN cell populations. (**D**) Frequencies and numbers of DN, DP, CD4$^+$ SP, and CD8$^+$ SP cells for each genotype. N = 11, 7, and 12 for *Cic$^{f/f}$*, *Cic$^{f/f}$;Cd4-Cre*, and *Cic$^{f/f}$;Vav1-Cre* mice, respectively. (**E**) Flow cytometric analysis of thymocytes isolated from 1-week-old *Cic$^{f/f}$* and *Cic$^{f/f}$;Vav1-Cre* mice using CD4 and CD8 markers. Total thymocyte numbers, and the frequencies and numbers of DN, DP, CD4$^+$ SP, and CD8$^+$ SP subsets in mice of each genotype, as well as representative plots are presented. N = 5 and 4 for *Cic$^{f/f}$* and *Cic$^{f/f}$;Vav1-Cre* mice, respectively. Data are representative of two independent experiments. Bar graphs represent the mean and SEM. *p < 0.05, **p < 0.01, ***p < 0.001, and ****p < 0.0001. One-way ANOVA with Tukey's multiple comparison test (**B–D**) and unpaired two-tailed Student's *t*-test (**E**) were used to calculate the corresponding p values. See also *Figure 1—source data 1*.

The online version of this article includes the following source data and figure supplement(s) for figure 1:

**Source data 1.** Raw data for *Figure 1*.

**Figure supplement 1.** Analysis of thymic γδT cells in *Cic$^{f/f}$;Vav1-Cre* mice.

**Figure supplement 2.** Analysis of splenic and thymic T cell subsets in 1-week-old *Cic$^{f/f}$;Vav1-Cre* mice.

**Figure supplement 3.** Flow cytometric analysis of recirculating CD4$^+$ T cells in the thymus of *Cic$^{f/f}$;Vav1-Cre* mice.

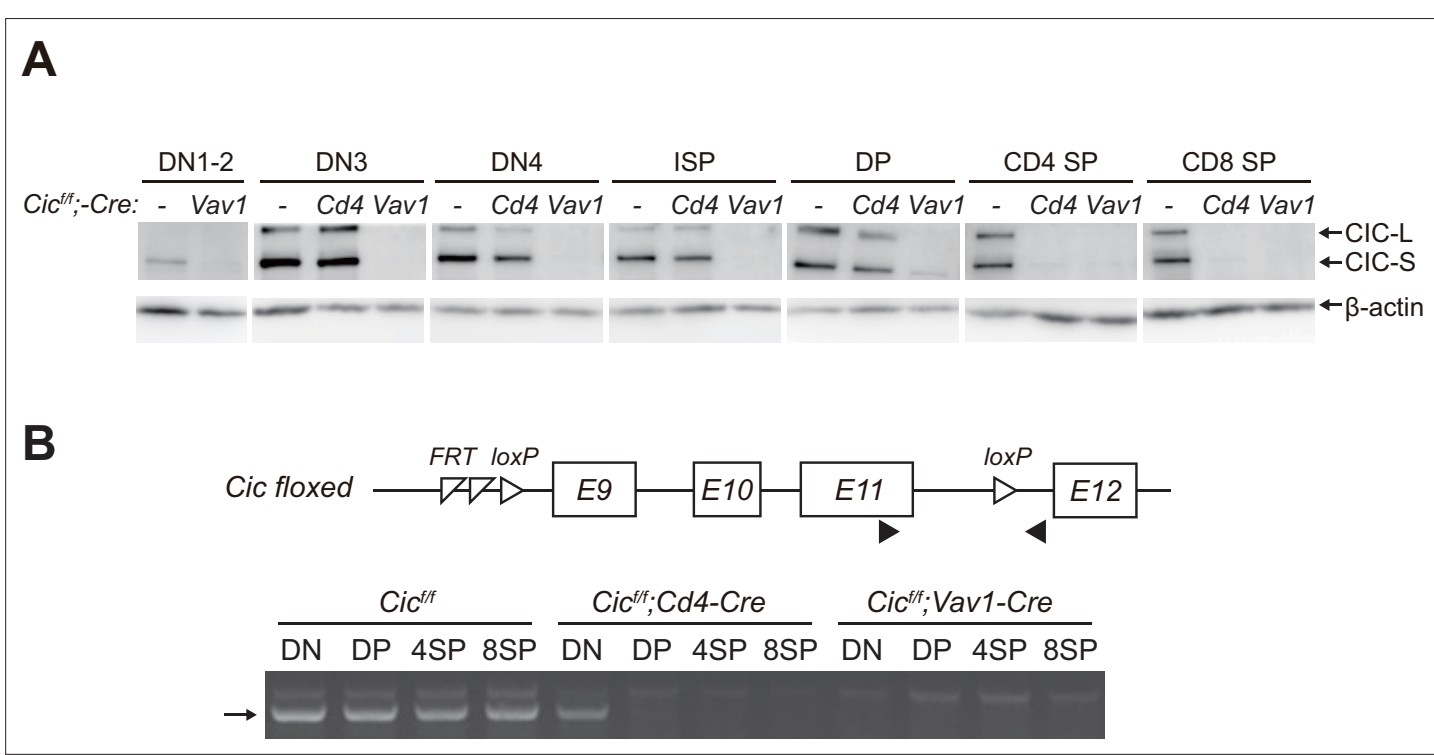

**Figure 2.** Stable capicua (CIC) expression in double-positive (DP) cells of *Cic$^{f/f}$;Cd4-Cre* mice. (**A**) Western blotting for detection of CIC levels in double-negative 1–2 (DN1-2), DN3, DN4, immature CD8$^+$ single-positive (ISP), double-positive (DP), CD4$^+$ SP, and CD8$^+$ SP cells from *Cic$^{f/f}$*, *Cic$^{f/f}$;Cd4-Cre*, and *Cic$^{f/f}$;Vav1-Cre* mice. Lin$^-$-gated DN1-2, DN3, DN4, ISP, DP, CD4$^+$ SP, and CD8$^+$ SP (TCRβ$^{hi}$) cells were sorted from mice of each genotype. (**B**) PCR analysis of *Cic* knock-out efficiency in Lin$^-$-gated DN, DP, CD4$^+$ SP, and CD8$^+$ SP (TCRβ$^{hi}$) cells from *Cic$^{f/f}$*, *Cic$^{f/f}$;Cd4-Cre*, and *Cic$^{f/f}$;Vav1-Cre* mice. Genomic DNA was extracted from sorted DN, DP, CD4$^+$ SP, and CD8$^+$ SP cells and subjected to PCR amplification of a part of the floxed *Cic* allele. Upper panel, schematic of the *Cic* floxed allele. Arrowheads indicate the primers used for PCR. Lower panel, representative agarose gel image of PCR products. The arrow indicates the PCR products corresponding to the amplified part of the floxed *Cic* allele. See also *Figure 2—source data 1*.

The online version of this article includes the following source data for figure 2:

**Source data 1.** Original and labelled files for western blot and PCR gel images.

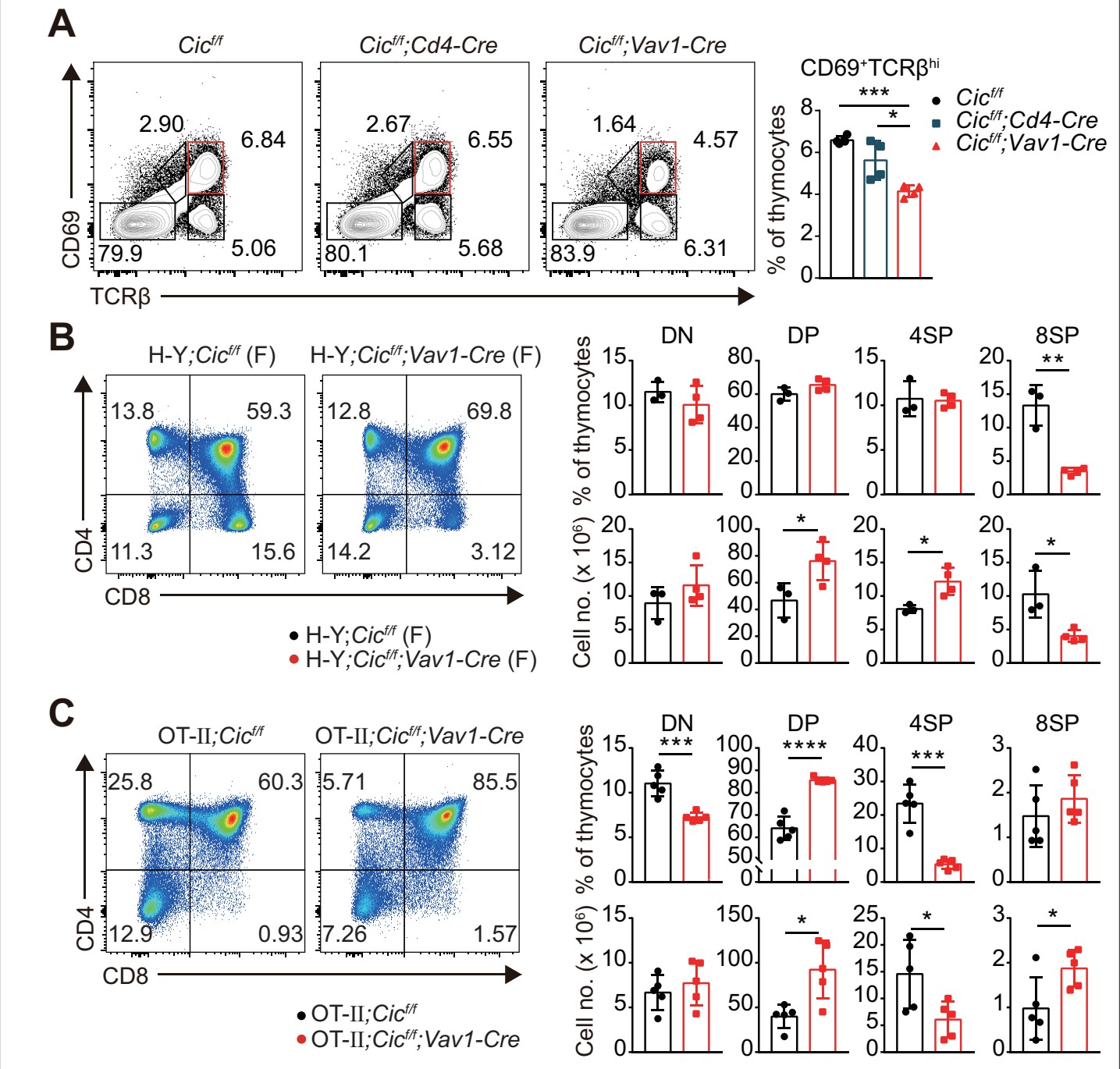

**Figure 3.** Defective positive selection in the absence of capicua (CIC).

The online version of this article includes the following source data and figure supplement(s) for figure 3:

**Source data 1.** Raw data for *Figure 3*.

**Figure supplement 1.** Analysis of H-Y TCR transgenic CD8⁺ T and immature CD8⁺ single positive (ISP) cells in female H-Y mice.

*Cic^{f/f};Vav1-Cre* mice for the expression of CD69 and TCRβ using flow cytometry. The frequency of CD69⁺TCRβ^{hi} cells, which represent post-positive-selection thymocytes (*Fu et al., 2009*), were significantly decreased in *Cic^{f/f};Vav1-Cre* mice compared with that of WT or *Cic^{f/f};Cd4-Cre* mice (*Figure 3A*). This reduction was also observed in the thymus of *Cic^{f/f};Vav1-Cre* mice at 1 week of age (*Figure 1—figure supplement 2C*). Next, we determined the effect of CIC deficiency on thymic positive selection

in mice expressing the MHC class I-restricted H-Y TCR transgene specific to the H-Y male antigen (*Kisielow et al., 1988*) or the MHC class II-restricted OT-II TCR transgene specific to ovalbumin (*Barnden et al., 1998*) at 7–9 weeks of age. Defects in SP thymocyte development were much more severe in TCR-transgenic hematopoietic lineage cell-specific *Cic*-null (H-Y;*Cic^f/f^;Vav1-Cre* or OT-II;*Cic^f/f^;Vav1-Cre*) mice than in non-transgenic polyclonal *Cic^f/f^;Vav1-Cre* mice: CIC deficiency dramatically reduced the populations of CD8+ and CD4+ SP cells in female H-Y TCR transgenic and OT-II TCR transgenic mice, respectively (*Figure 3B and C*, *Figure 3—figure supplement 1A*). The frequency of CD24^lo^TCRβ^hi^ ISP cells among total CD8+ SP thymocytes was significantly increased in female H-Y;*Cic^f/f^;Vav1-Cre* mice (*Figure 3—figure supplement 1B*), excluding the possibility that the decreased frequency of CD8+ SP thymocytes could result from decreased ISP cell formation in female H-Y;*Cic^f/f^;Vav1-Cre* mice. Taken together, these results indicate that CIC controls the positive selection of thymocytes.

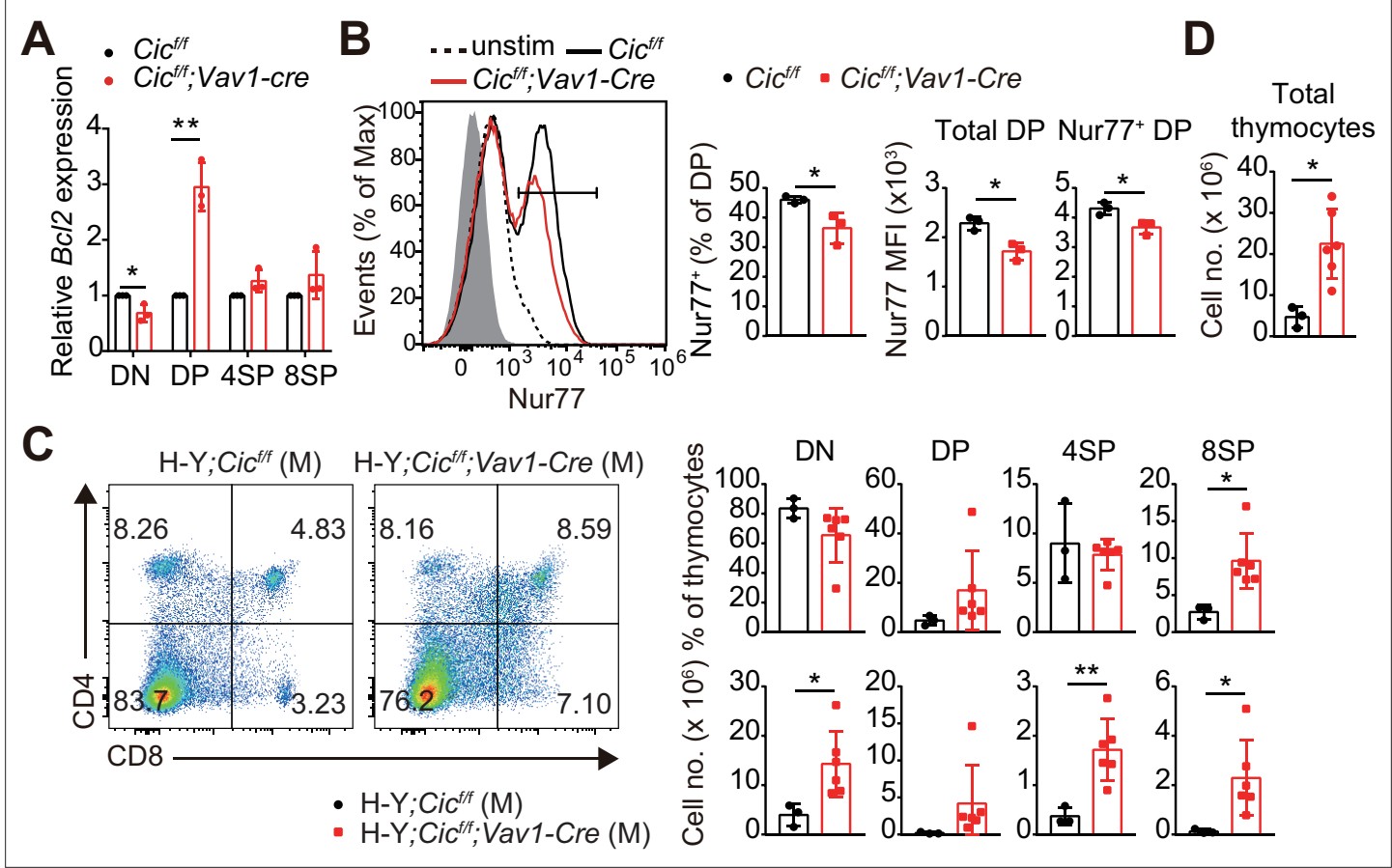

**Figure 4.** Defective negative selection in the absence of capicua (CIC). (**A**) qRT-PCR quantification of *Bcl2* expression in double-negative (DN), double-positive (DP), CD4+ single-positive (4SP), and CD8+ SP (8SP, TCRβ^hi^) cells of *Cic^f/f^* and *Cic^f/f^;Vav1-Cre* mice. N = 3 for each group. (**B**) Flow cytometric analysis of Nur77 expression in DP thymocytes from *Cic^f/f^* and *Cic^f/f^;Vav1-Cre* mice. Freshly isolated thymocytes were treated with plate-coated anti-CD3 (5 µg/ml) and anti-CD28 (10 µg/ml) antibodies for 2 hr and subsequently subjected to flow cytometry. Representative histograms for Nur77 expression in DP thymocytes of *Cic^f/f^* (black line) and *Cic^f/f^;Vav1-Cre* (red line) mice overlaid with isotype (gray shaded) and unstimulated (dotted line) control histograms (left), the frequency of Nur77+ DP cells (middle), and Nur77-derived mean fluorescence intensities (MFIs) of total and Nur77+ DP cells (right) are presented. N = 3 for each genotype. Data are representative of two independent experiments. (**C and D**) Thymocytes from male H-Y;*Cic^f/f^* (N = 3) and H-Y;*Cic^f/f^;Vav1-Cre* (N = 6) mice were analyzed for CD4 and CD8 expression. (**C**) Representative flow cytometry plots (left) and frequencies (top, right) and numbers (bottom, right) of DN, DP, CD4+ SP (4SP), and CD8+ SP (8SP) cells are shown. (**D**) Total thymocyte numbers. Data are representative of two independent experiments. Bar graphs represent the mean and SEM. *p < 0.05 and **p < 0.01. Unpaired two-tailed Student's *t*-test was used to calculate the corresponding p values. See also *Figure 4—source data 1*.

The online version of this article includes the following source data and figure supplement(s) for figure 4:

**Source data 1.** Raw data for *Figure 4*.

**Figure supplement 1.** Analysis of H-Y TCR transgenic CD8+ T and immature CD8+ single positive (ISP) cells in male H-Y mice.

**Figure supplement 2.** Analysis of TCR Vβ chains in CIC-deficient CD4+ SP thymocytes.

## Impaired thymic negative selection in *Cic^f/f^;Vav1-Cre* mice

Most thymocytes expressing autoreactive TCRs are eliminated through negative selection during the DP and SP developmental stages (*Klein et al., 2014*). The strong TCR signal induced by the interaction of a self-reactive TCR with the self-pMHC molecule triggers apoptosis of autoreactive T cells (*Hogquist and Jameson, 2014*). Interestingly, we found that the expression of *Bcl2*, a representative anti-apoptotic gene (*Vaux et al., 1988*) and an inhibitor of negative selection (*Williams et al., 1998*), was markedly increased in DP cells but not in DN and SP cells from *Cic^f/f^;Vav1-Cre* mice (*Figure 4A*). Furthermore, the TCR stimulation-induced expression of Nur77, an orphan receptor that promotes apoptosis and negative selection of thymocytes (*Calnan et al., 1995*), was significantly reduced in DP cells from *Cic^f/f^;Vav1-Cre* mice compared to that in WT mice (*Figure 4B*). To determine whether CIC regulates thymic negative selection, we analyzed thymocytes from male H-Y;*Cic^f/f^* and H-Y;*Cic^f/f^;Vav1-Cre* mice by flow cytometry. As previously reported (*Kisielow et al., 1988*), massive negative selection of DP and CD8^+^ SP thymocytes was observed in male H-Y TCR transgenic mice (*Figure 4C*), in which the male-specific H-Y autoantigens are expressed in thymic antigen-presenting cells. The CD8^+^ SP population and total thymic cell number significantly increased in male H-Y;*Cic^f/f^;Vav1-Cre* mice compared to male H-Y;*Cic^f/f^* mice (*Figure 4C and D*, *Figure 4—figure supplement 1A*), whereas the frequency of CD24^lo^TCRβ^hi^ ISP cells among all CD8^+^ SP thymocytes, which were less than 10%, was comparable between male H-Y;*Cic^f/f^* and H-Y;*Cic^f/f^;Vav1-Cre* mice (*Figure 4—figure supplement 1B*). Overall, our data demonstrate that CIC is required for negative selection mediated by TCR activation-induced apoptosis.

## Changes in the frequency of the variable (V) segments of the TCRβ chain in CIC-deficient CD4^+^ SP thymocytes

Developing T cells mature into SP cells through thymic selection processes. Consequently, numerous T cells with various TCRs constitute the TCR repertoire. As positive and negative selection determine the fate of T cells based on their TCR, defects in these processes can lead to changes in the TCR repertoire (*Lu et al., 2019*; *Martínez-Riaño et al., 2019*). To find out whether CIC deficiency alters the TCR repertoire of SP thymocytes, we analyzed the frequency of the variable (V) segments of the TCRβ (Vβ) chain on CD4^+^ non-Treg (CD4^+^CD25^-^Foxp3^-^) and Treg (CD4^+^CD25^+^Foxp3^+^) cells from WT and *Cic^f/f^;Vav1-Cre* mice by flow cytometry using antibodies against 10 different Vβ chains. Among the 10 Vβ chains tested, the frequency of CD4^+^ non-Treg cells expressing eight different Vβ chains was significantly changed: the frequency of Vβ5.1/5.2, 9, 11, 12, and 13 chain-expressing CD4^+^ non-Treg cells was decreased in *Cic^f/f^;Vav1-Cre* mice, whereas that of Vβ 6, 7, and 8.1/8.2 chain-expressing CD4^+^ non-Treg cells was increased (*Figure 4—figure supplement 2A*). In the Treg cell compartment, the frequencies of Vβ7 and Vβ9 were significantly changed in *Cic^f/f^;Vav1-Cre* mice compared to WT mice (*Figure 4—figure supplement 2B*). These data suggest that impaired thymic selection due to CIC deficiency might change the TCR repertoire of thymic SP cell populations.

## Attenuated TCR signaling in CIC-deficient DP thymocytes

Because both positive and negative selection are mediated by TCR signaling strength (*Klein et al., 2014*), we investigated whether CIC regulates the TCR signaling pathway. First, we analyzed the surface expression of CD5, which is strongly correlated with TCR signaling intensity (*Azzam et al., 1998*), in DP and SP thymocytes from WT, *Cic^f/f^;Cd4-Cre,* and *Cic^f/f^;Vav1-Cre* mice. CD5 levels were substantially decreased in DP cells from *Cic^f/f^;Vav1-Cre* mice compared to those from WT or *Cic^f/f^;Cd4-Cre* mice, whereas this change was subtle in the CD4^+^ SP cell population and absent in the CD8^+^ SP cell population (*Figure 5A*). Consistent with previous results (*Figure 3A*, *Figure 1—figure supplement 2C*), the frequency of CD69^+^ post-selection DP thymocytes was significantly decreased in *Cic^f/f^;Vav1-Cre* mice (*Figure 5B*). Decreased CD5 expression was mostly observed in CD69^-^ pre-selection DP thymocytes from *Cic^f/f^;Vav1-Cre* mice (*Figure 5C*). These results were recapitulated in TCR transgenic mice (*Figure 5—figure supplement 1*). Next, we examined the activation of the TCR signaling pathway in thymocytes from WT, *Cic^f/f^;Cd4-Cre,* and *Cic^f/f^;Vav1-Cre* mice upon TCR stimulation. As with CD5 levels, calcium influx sharply decreased in DP cells from *Cic^f/f^;Vav1-Cre* mice, but not in SP thymocytes (*Figure 5D*). We also observed a moderate decrease in calcium influx in DP cells from *Cic^f/f^;Cd4-Cre* mice (*Figure 5D*), suggesting that CIC sensitively regulates TCR activation-induced calcium influx in DP thymocytes. To further evaluate CIC regulation of TCR signaling in DP

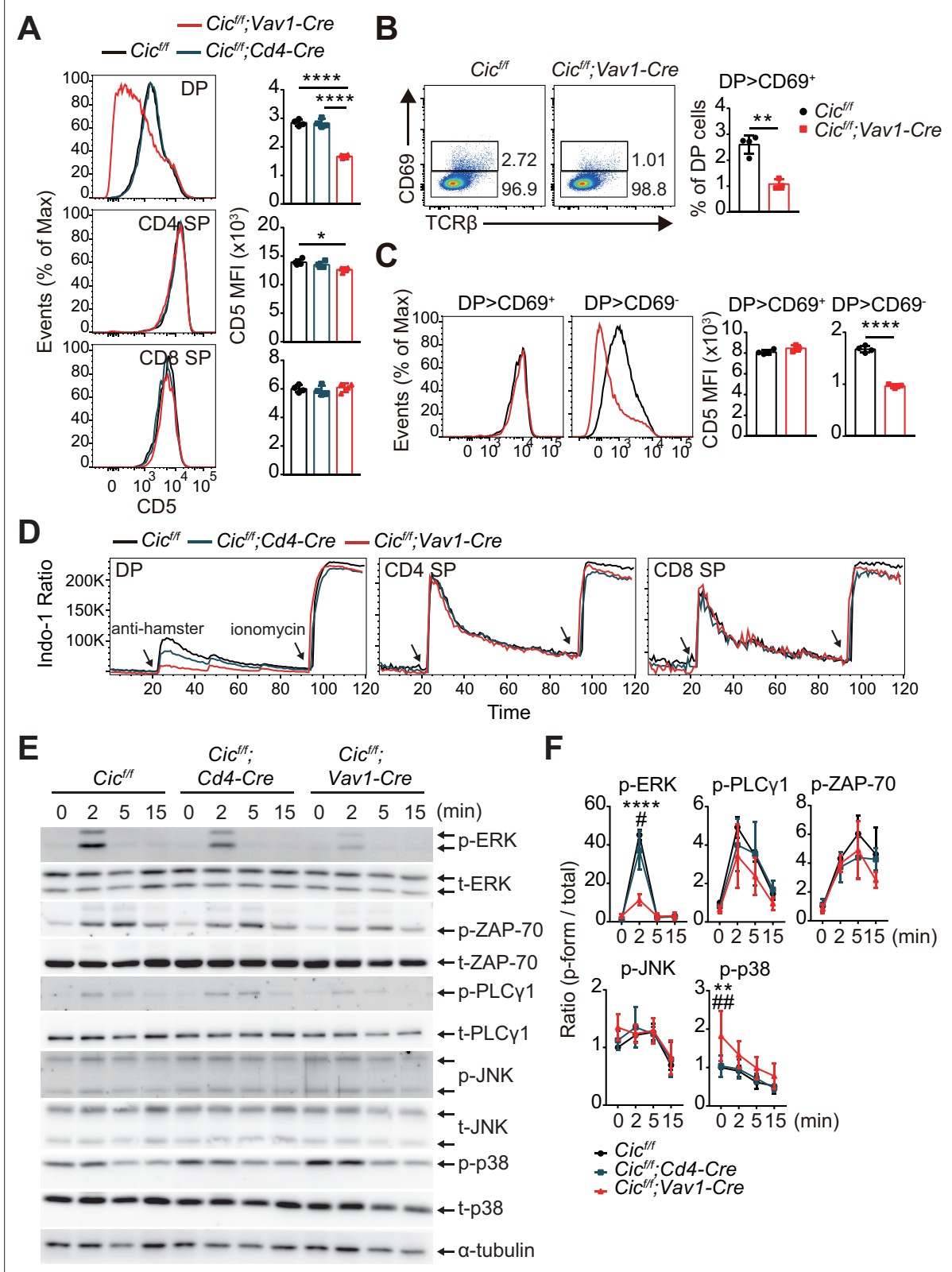

**Figure 5.** Attenuated TCR signaling in capicua (CIC)-deficient double-positive (DP) thymocytes. (**A**) Thymocytes from 7-week-old $Cic^{f/f}$, $Cic^{f/f};Cd4$-Cre, and $Cic^{f/f};Vav1$-Cre mice were FACS-gated into the DP, CD4+ single-positive (SP), and CD8+ SP (TCRβ$^{hi}$) cell population, and the CD5 mean fluorescence intensity (MFI) was measured. Representative histograms of CD5 expression in each cell population (left) and calculated CD5 MFI values (right) are shown (N = 4 per group). (**B and C**) DP cells from $Cic^{f/f}$ (N = 4) and $Cic^{f/f};Vav1$-Cre (N = 3) mice were analyzed for CD69 and CD5 expression.

*Figure 5 continued on next page*

*Figure 5 continued*

(**B**) Representative flow cytometry plots (left) and frequencies (right) of CD69$^+$ DP cells are shown. (**C**) Representative histograms of CD5 expression in CD69$^+$ or CD69$^-$ DP cells (left) and CD5 MFI values in each cell population (right) are shown. (**D**) TCR stimulation-induced Ca$^{2+}$ influx in DP, CD4$^+$ SP, and CD8$^+$ SP (TCRβ$^{hi}$) thymocytes from 7-week-old *Cic$^{f/f}$*, *Cic$^{f/f}$;Cd4-Cre*, and *Cic$^{f/f}$;Vav1-Cre* mice. Data are representative of at least three independent experiments. (**E and F**) Western blot analysis of TCR cascade component activation in DP thymocytes from 7-week-old *Cic$^{f/f}$*, *Cic$^{f/f}$;Cd4-Cre*, and *Cic$^{f/f}$;Vav1-Cre* mice. Sorted DP cells were stimulated with soluble anti-CD3 and anti-CD4 antibodies for the times indicated. (**E**) Representative western blot images are shown. (**F**) Signal densities were measured using ImageJ software and are presented as ratios of phosphorylated to total forms for each protein (N = 3). Graphs represent the mean and SEM. *p < 0.05, **p < 0.01, ***p < 0.001, and ****p < 0.0001. #p < 0.05 and ##p < 0.01 (comparison between *Cic$^{f/f}$;Cd4-Cre* and *Cic$^{f/f}$;Vav1-Cre* mice). One-way (**A**) or two-way (**F**) ANOVA with Tukey's multiple comparison test and unpaired two-tailed Student's *t*-test (**B and C**) were used to calculate the corresponding p values. See also *Figure 5—source data 1*, *Figure 5—source data 2*.

The online version of this article includes the following source data and figure supplement(s) for figure 5:

**Source data 1.** Raw data for *Figure 5*.

**Source data 2.** Original and labeled files for western blot images.

**Figure supplement 1.** Analysis of CD5 expression in TCR transgenic thymocytes.

thymocytes, we assessed the activation of key TCR signaling cascade components in DP cells of WT, *Cic$^{f/f}$;Cd4-Cre*, and *Cic$^{f/f}$;Vav1-Cre* mice after TCR stimulation by treatment with anti-CD3 and anti-CD4. Among the components tested, including ZAP-70, PLCγ, ERK, JNK, and p38, phospho-ERK levels were significantly decreased in DP cells from *Cic$^{f/f}$;Vav1-Cre* mice compared to those from WT and *Cic$^{f/f}$;Cd4-Cre* mice (*Figure 5E and F*). Together, these results demonstrate that CIC deficiency attenuates TCR signaling by inhibiting calcium influx and ERK activation, especially in DP thymocytes.

## Abnormal thymic T cell development and reduced TCR signaling intensity in *Cic$^{f/f}$;Vav1-Cre* mice are T cell-intrinsic

Although most of the cells that make up thymocytes are developing T cells, other types of immune cells, including B cells and dendritic cells, also exist in small proportions and participate in the regulation of T cell development (*Klein et al., 2014*). Because CIC expression was abolished in whole immune cells of *Cic$^{f/f}$;Vav1-Cre* mice, it is unclear whether abnormal thymic T cell development and decreased TCR signaling in *Cic$^{f/f}$;Vav1-Cre* mice were T cell-intrinsic or -extrinsic. To clarify this issue, we generated mixed BM chimeric mice by transferring the same number of BM cells from Thy1.1/Thy1.2 heterozygous WT and Thy1.1/Thy1.1 homozygous *Cic$^{f/f}$;Vav1-Cre* mice into irradiated Thy1.2/Thy1.2 homozygous WT recipient mice (mixed WT:*Cic$^{f/f}$;Vav1-Cre* BM chimera), and analyzed thymic T cells in the chimeras after 8 weeks of reconstitution (*Figure 6A*). Similar to the observations in *Cic$^{f/f}$;Vav1-Cre* mice (*Figure 1C–E*, *Figure 1—figure supplement 2B*), the frequency of DN4 and CD4$^+$ SP cells was significantly lower in the CIC-deficient cell compartment than in the WT cell counterpart in the same mixed WT:*Cic$^{f/f}$;Vav1-Cre* BM chimeric mice (*Figure 6B and C*). The frequency of CD69$^+$TCRβ$^{hi}$ cells was also significantly lower in the CIC-deficient cell compartment than in the WT cell compartment (*Figure 6D*), indicating that the impaired positive selection in *Cic$^{f/f}$;Vav1-Cre* mice was T cell-intrinsic. Furthermore, similar to the results in *Figure 5A*, CD5 levels were most strongly reduced in DP thymocytes derived from *Cic$^{f/f}$;Vav1-Cre* BM cells (*Figure 6E*).

As an alternative approach to examine T cell intrinsic function of CIC in the regulation of thymic T cell development and selection processes, we generated and analyzed the proximal *Lck* promoter (p*Lck*)-driven *Cre*-mediated T cell-specific *Cic* null (*Cic$^{f/f}$;pLck-Cre*) mice. Although p*Lck-Cre* is expressed during the DN2 cell stage (*Lee et al., 2001*), CIC was substantially expressed in DN3 and DN4 cells from *Cic$^{f/f}$;pLck-Cre* mice (*Figure 6—figure supplement 1A*). Moreover, a small amount of CIC still existed in DP thymocytes from *Cic$^{f/f}$;pLck-Cre* mice (*Figure 6—figure supplement 1A*). The frequency of DN thymocytes was comparable between WT and *Cic$^{f/f}$;pLck-Cre* mice at 7 weeks of age (*Figure 6—figure supplement 1B*), concurrent with the expression of a substantial amount of CIC in DN3 and DN4 cells from *Cic$^{f/f}$;pLck-Cre* mice (*Figure 6—figure supplement 1A*). Although there was no significant difference observed in the frequencies of DP and SP subsets between WT and *Cic$^{f/f}$;pLck-Cre* mice, there was a tendency for a slight increase in the frequency of DN and DP thymocytes and a decrease in the frequency of CD4$^+$ SP cells (*Figure 6—figure supplement 1C*), similar to those in *Cic$^{f/f}$;Vav1-Cre* mice at 7 weeks of age (*Figure 1D*). Furthermore, the frequency of CD69$^+$TCRβ$^{hi}$ cells and CD5 levels in DP thymocytes were significantly decreased in *Cic$^{f/f}$;pLck-Cre* mice compared to WT mice. However, the fold decrease was subtle in *Cic$^{f/f}$;pLck-Cre* mice than in *Cic$^{f/f}$;Vav1-Cre* mice

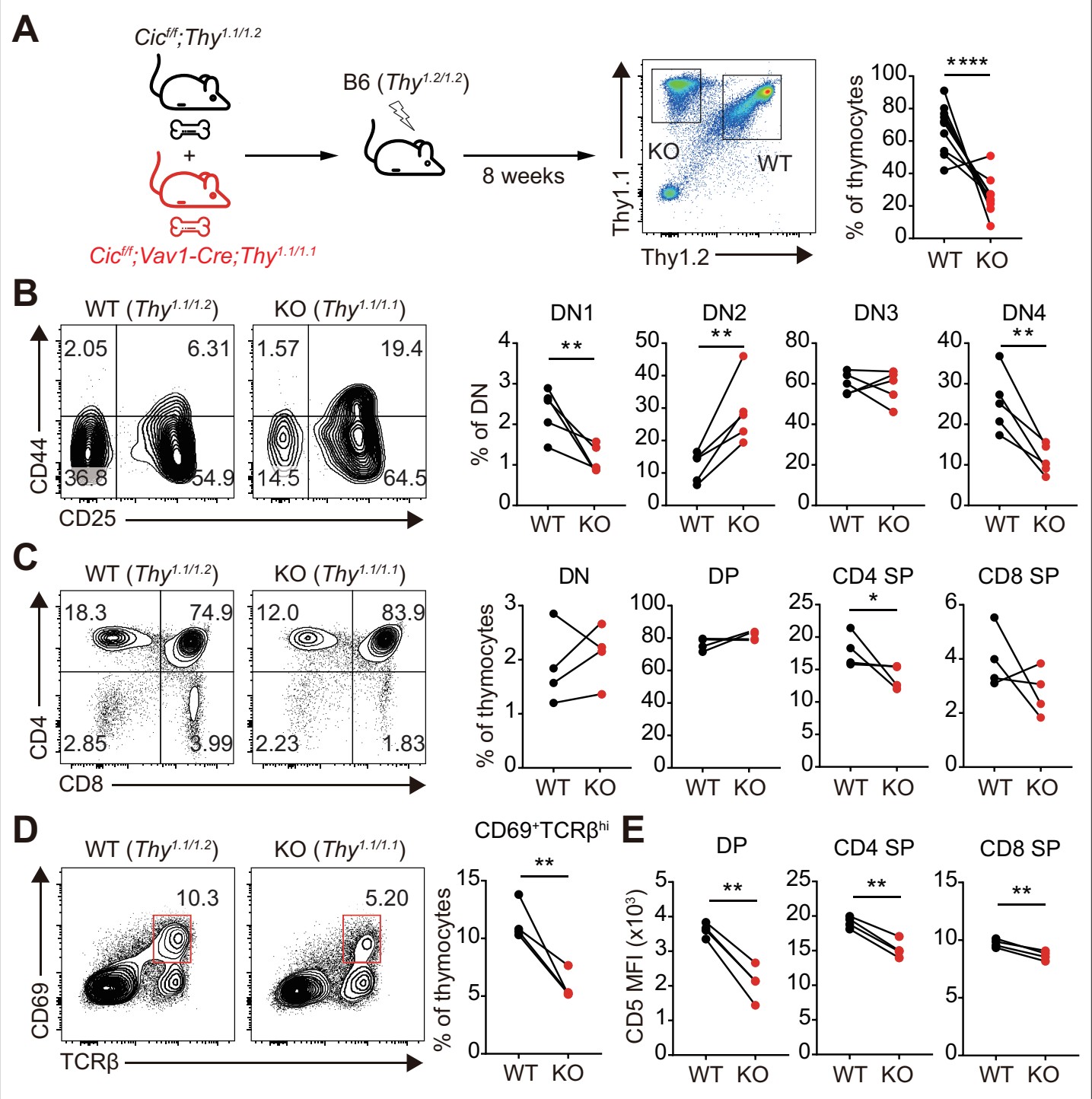

**Figure 6.** Altered T cell development and TCR intensity in *Cic^f/f;Vav1-Cre* mice are caused by CIC loss in T cells. (**A**) Schematic of the generation and analysis of mixed bone marrow (BM) chimeric mice. Equal numbers of BM cells from *Cic^f/f;Thy^1.1/1.2* (WT) and *Cic^f/f;Vav1-Cre;Thy^1.1/1.1* (KO) mice were mixed and transferred to irradiated B6 (*Thy^1.2/1.2*) recipient mice. Representative FACS plot showing the thymocytes of different origin (left) and their frequencies (right) are presented (N = 10). (**B–D**) Flow cytometric analysis of thymocytes from mixed BM chimeras (N = 4) for the frequencies of (**B**) double-negative (DN) subsets based on CD44 and CD25 expression, (**C**) DN, double-positive (DP), CD4⁺ single-positive (SP), and CD8⁺ SP cells, and (**D**) post-positive selection subsets (CD69⁺TCRβ^hi). The CD69⁺TCRβ^hi cell population is highlighted by the red box in the flow cytometry plots in (**D**). (**E**) Flow cytometric analysis of surface expression levels of CD5 in DP, CD4⁺ SP, and CD8⁺ SP (TCRβ^hi) thymocytes derived from WT and KO BM cells in the same BM chimeric mice (N = 4). Data are representative of two independent experiments. Graphs represent the mean and SEM. *p < 0.05, **p < 0.01, and ****p < 0.0001. Unpaired two-tailed Student's *t*-test was used to calculate the corresponding p values. See also *Figure 6—source data 1*.

*Figure 6 continued on next page*

*Figure 6 continued*

The online version of this article includes the following source data and figure supplement(s) for figure 6:

**Source data 1.** Raw data for *Figure 6*.

**Figure supplement 1.** Analysis of thymocytes in *Cic^f/f;pLck-Cre* mice.

(*Figure 6—figure supplement 1D and E*). Overall, *Cic^f/f;pLck-Cre* mice exhibited milder defects in positive selection and TCR signaling in DP thymocytes compared to *Cic^f/f;Vav1-Cre* mice, which was probably due to the incomplete removal of CIC expression in DP thymocytes of *Cic^f/f;pLck-Cre* mice (*Figure 6—figure supplement 1A*). Taken together, these data suggest that defects in thymic T cell development and TCR signaling in *Cic^f/f;Vav1-Cre* mice are T cell-intrinsic.

## Identification of CIC target genes responsible for attenuated TCR signaling in CIC-deficient DP thymocytes

To understand how CIC regulates the thymic selection process and TCR signaling in DP cells at the molecular level, we analyzed the gene expression profiles of DP thymocytes from WT and *Cic^f/f;Vav1-Cre* mice by RNA sequencing. A total of 482 differentially expressed genes (DEGs; fold change >2 and p-value < 0.05), including 263 upregulated and 219 downregulated genes, were identified in CIC-deficient DP cells (*Supplementary file 1*). Gene Ontology (GO) analysis revealed that genes involved in the inactivation of MAPKs and anti-apoptotic processes were significantly enriched among the upregulated DEGs (*Supplementary file 2*), consistent with the characteristics found in CIC-deficient DP thymocytes. We also identified several known CIC target genes among the DEGs, including *Etv1*, *Etv4*, *Etv5*, *Spry4*, *Dusp6*, *Dusp4,* and *Spred1* (*Fryer et al., 2011*; *Weissmann et al., 2018*; *Yang et al., 2017*; *Figure 7A* and *Supplementary file 1*). Of these, *Spry4*, *Dusp4*, *Dusp6,* and *Spred1* are of particular interest because they are negative regulators of ERK activation (*Kidger and Keyse, 2016*; *Sasaki et al., 2003*; *Wakioka et al., 2001*). Murine SPRY4 suppresses Ras-independent ERK activation by binding to Raf1 (*Sasaki et al., 2003*), whereas it represses the insulin receptor and EGFR-induced ERK signaling upstream of Ras in humans (*Leeksma et al., 2002*). SPRED1 also inhibits the activation of ERK by suppressing Raf activation (*Wakioka et al., 2001*). DUSP6 specifically dephosphorylates activated ERK1/2, whereas DUSP4 acts on ERK and other MAPKs, such as JNK and p-38 (*Caunt and Keyse, 2013*). Additionally, SPRY4 suppresses calcium mobilization in HEK293T cells by inhibiting phosphatidylinositol 4,5-biphosphate ($PIP_2$) hydrolysis induced by VEGF-A without affecting PLCγ phosphorylation (*Ayada et al., 2009*). Thus, we investigated the association between derepression of *Spry4*, *Dusp4*, *Dusp6,* and *Spred1* and attenuated TCR signaling in CIC-deficient DP thymocytes. First, we measured the levels of *Spry4*, *Dusp4*, *Dusp6,* and *Spred1* in DN, DP, CD4^+ SP, and CD8^+ SP thymocytes from WT and *Cic^f/f;Vav1-Cre* mice by qRT-PCR. The expression of all four genes was significantly upregulated in DP cells from *Cic^f/f;Vav1-Cre* mice relative to those from WT mice (*Figure 7B*), verifying the RNA sequencing data (*Supplementary file 1* and *Figure 7A*). The expression of all four genes was also upregulated in CD4^+ and CD8^+ SP thymocytes from *Cic^f/f;Vav1-Cre* mice; *Spry4* and *Dusp6* were more dramatically upregulated in DP cells than in SP cells (*Figure 7B*). Considering the different gene expression changes in each cell type, as well as the previously known functional significance in the regulation of ERK activation and calcium influx, two of the four CIC target genes, *Spry4* and *Dusp6*, were selected for further investigation of their effect on TCR signaling in DP thymocytes.

We infected thymocytes with retroviruses expressing *Spry4* or *Dusp6* and analyzed phospho-ERK and CD5 levels and calcium influx by flow cytometry. The levels of phospho-ERK were greatly decreased in both SRPY4- and DUSP6-overexpressing DP thymocytes treated with anti-CD3 (*Figure 7C*), confirming their inhibitory effect on ERK activation (*Groom et al., 1996*; *Muda et al., 1996*; *Sasaki et al., 2003*). In contrast, calcium influx induced by TCR stimulation was almost completely suppressed in DP thymocytes by SPRY4 overexpression, but not by DUSP6 overexpression (*Figure 7D*). Moreover, CD5 expression was substantially reduced in SRPY4-overexpressing cells, but not in DUSP6-overexpressing cells (*Figure 7E*). These data imply that *Spry4* derepression might critically contribute to attenuated TCR signaling in CIC-deficient DP thymocytes.

Finally, to determine whether impaired thymic selection and TCR signaling in *Cic^f/f;Vav1-Cre* mice were caused by derepression of *Spry4*, we generated *Cic* and *Spry4* double mutant (*Spry4^-/-;Cic^f/f;Vav1-Cre*) mice and analyzed thymic T cells in WT, *Cic^f/f;Vav1-Cre*, and *Spry4^-/-;Cic^f/f;Vav1-Cre* mice at

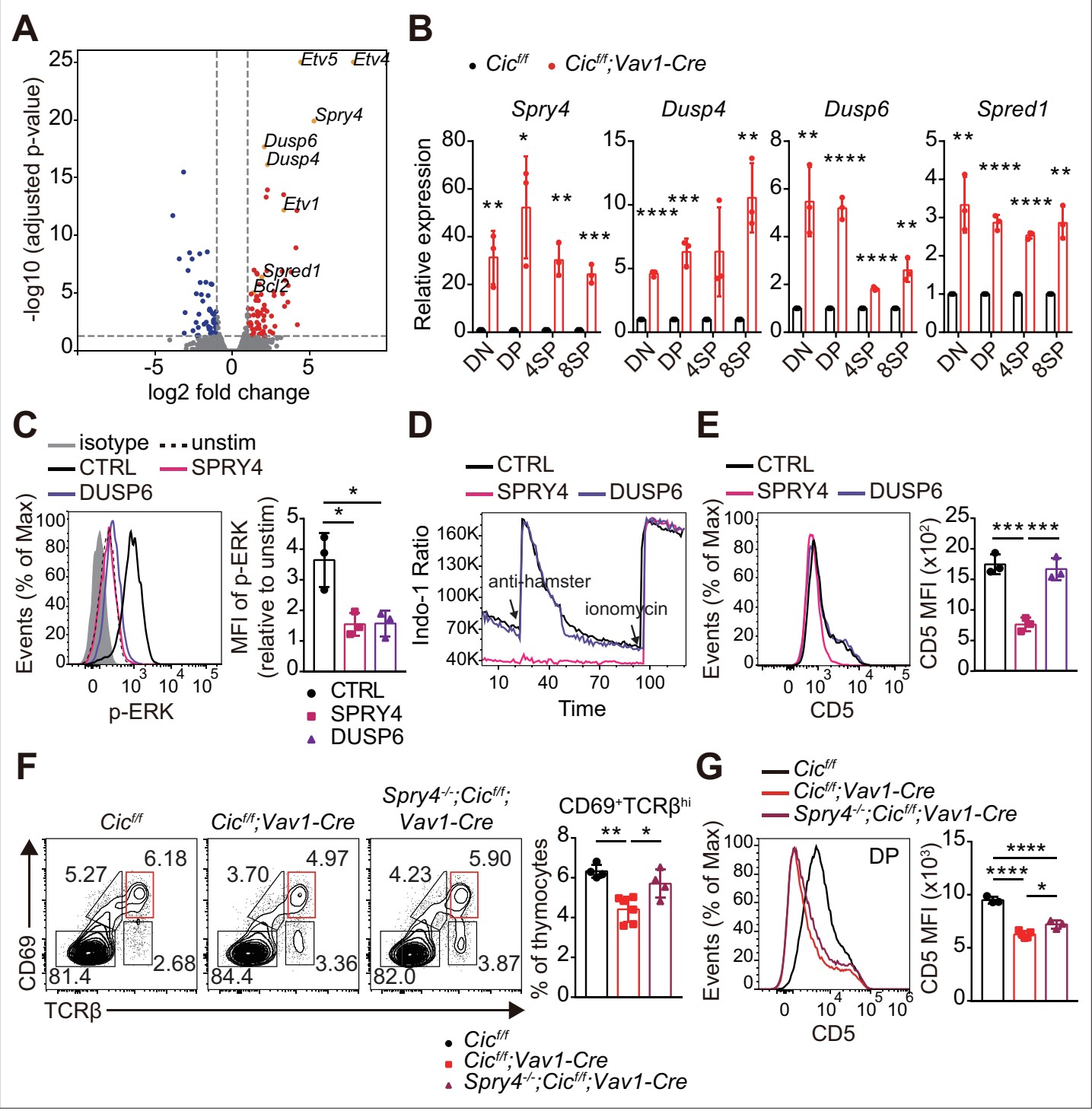

**Figure 7.** Identification of capicua (CIC) target genes regulating TCR signaling in double-positive (DP) thymocytes. (**A**) Volcano plot showing differentially expressed genes (DEGs) in CIC-deficient DP thymocytes (fold change, > 2; adjusted P value, < 0.05). CIC target genes and *Bcl2* are indicated at the corresponding dots. (**B**) qRT-PCR quantification of *Spry4*, *Dusp4*, *Dusp6*, and *Spred1* expression in double-negative (DN), DP, CD4+ single-positive (4SP), and CD8+ SP (8SP) thymocytes from *Cic^f/f* and *Cic^f/f;Vav1-Cre* mice. N = 3 for each group. (**C–E**) Effects of SPRY4 and DUSP6 overexpression on TCR signaling in DP cells. Thymocytes were infected with retroviruses co-expressing GFP and either SPRY4 or DUSP6, and subjected to flow cytometry for (**C**) ERK activation, (**D**) Ca²⁺ influx, and (**E**) CD5 expression in GFP+ DP thymocytes. Three independent experiments were performed. (**F**) Thymocytes from 7-week-old *Cic^f/f*, *Cic^f/f;Vav1-Cre*, and *Spry4^-/-;Cic^f/f;Vav1-Cre* mice were analyzed for surface expression of CD69 and TCRβ. Representative FACS plots (left) and the frequency of CD69+TCRβ^hi cells (right) are shown. The CD69+TCRβ^hi cell population is highlighted by the red box in the FACS plots. N = 4, 6, and four for *Cic^f/f*, *Cic^f/f;Vav1-Cre*, and *Spry4^-/-;Cic^f/f;Vav1-Cre* mice, respectively. (**G**) CD5 levels in DP thymocytes

*Figure 7 continued on next page*

*Figure 7 continued*

from mice used in (**F**). Representative FACS plots (left) and CD5 mean fluorescence intensities (MFIs; right) are shown. N = 3, 5, and three for *Cic*[f/f], *Cic*[f/f];*Vav1-Cre*, and *Spry4*[-/-];*Cic*[f/f];*Vav1-Cre* mice, respectively. Bar graphs represent the mean and SEM. *p < 0.05, **p < 0.01, ***p < 0.001, and ****p < 0.0001. Unpaired two-tailed Student's *t*-test (**B**) and one-way ANOVA with Tukey's multiple comparison test (**C, E, F** and **G**) were used to calculate the corresponding p values. See also *Figure 7—source data 1*.

The online version of this article includes the following source data and figure supplement(s) for figure 7:

**Source data 1.** Raw data for *Figure 7*.

**Figure supplement 1.** Analysis of T cell subsets in the thymus and spleen of *Spry4*[-/-];*Cic*[f/f];*Vav1-Cre* mice.

7–8 weeks of age. The decreased frequency of CD69[+]TCRβ[hi] cells and CD5 levels of DP thymocytes in *Cic*[f/f];*Vav1-Cre* mice were significantly, but only partially, rescued in *Spry4*[-/-];*Cic*[f/f];*Vav1-Cre* mice (*Figure 7F and G*), suggesting a partial recovery of the CIC deficiency-mediated defective positive selection process by loss of SPRY4. However, the decreased DN4 cell frequency and Nur77 expression in DP thymocytes were not rescued in *Spry4*[-/-];*Cic*[f/f];*Vav1-Cre* mice (*Figure 7—figure supplement 1A and B*), indicating that *Spry4* derepression alone is not sufficient to induce abnormal DN cell development in *Cic*[f/f];*Vav1-Cre* mice and suppressed TCR activation-induced apoptosis in CIC-deficient DP thymocytes. Taken together, our findings demonstrate that CIC tightly controls thymic T cell development and TCR signaling by repressing multiple CIC target genes, including *Dusp4*, *Dusp6*, *Spred1*, and *Spry4*, in a cell type- and/or developmental stage-specific manner.

## Discussion

Our study uncovered the role of CIC in thymic T cell development via in-depth analyses of thymocytes in various CIC-deficient mouse models. CIC deficiency partially suppressed the DN3-to-DN4 transition. However, the decreased DN4 cell population did not prevent the formation of thymic DP cells in *Cic*[f/f];*Vav1-Cre* mice. Thus, this defect was not sufficient to significantly affect the transition from the DN to DP stage of thymic T cell development in *Cic*[f/f];*Vav1-Cre* mice. Interestingly, similar to *Cic*[f/f];*Vav1-Cre* mice, ERK-deficient mice also exhibit a partial block in DN3-to-DN4 maturation without defects in maturation to the DP stage of development (*Fischer et al., 2005*). Because TCR stimulation-induced ERK activation was markedly suppressed in CIC-deficient DP thymocytes (*Figure 5E and F*), we inferred that the formation of the abnormal DN subset in the thymus of *Cic*[f/f];*Vav1-Cre* mice was, at least in part, caused by reduced ERK activity in DN cells deficient in CIC. Consistent with this inference, the *Spry4*, *Dusp4*, *Dusp6*, and *Spred1* genes, involved in the inhibition of ERK activation (*Kidger and Keyse, 2016*; *Sasaki et al., 2003*; *Wakioka et al., 2001*), were significantly derepressed in CIC-deficient DN cells (*Figure 7B*). Further study on how CIC controls pre-TCR signaling is critical to better understand CIC regulation of thymic T cell development during the DN stage.

Our previous study showed that the frequency of thymic CD4[+] and CD8[+] SP cells was comparable between WT and *Cic*[f/f];*Vav1-Cre* mice at 9 weeks of age (*Park et al., 2017*). However, thymic T cell subset analysis at a younger age revealed a significant decrease in the frequency of SP thymocytes in *Cic*[f/f];*Vav1-Cre* mice (*Figure 1D and E*). This phenotypic change was not due to the enhanced accumulation of circulating peripheral T cells in the thymus of *Cic*[f/f];*Vav1-Cre* mice at 9 weeks of age (*Figure 1—figure supplement 3B*). This difference could be explained by the differential effects of positive and negative selection on the formation of SP thymocytes during ontogeny. It is known that negative selection is inefficient early in ontogeny and becomes more efficient with age (*He et al., 2013*), implying that the process of positive selection might predominantly determine the size of the thymic SP cell population during neonatal life. Since *Cic*[f/f];*Vav1-Cre* mice display defects in both positive and negative selection, it is conceivable that the decreased frequency of SP thymocytes in 1-week-old *Cic*[f/f];*Vav1-Cre* mice were caused by a defect in positive selection, which was attenuated by ineffective negative selection at 7 weeks of age or older.

CIC deficiency, especially in DP thymocytes, disrupts the positive and negative selection of thymocytes, as evidenced by the impaired thymic selection process in *Cic*[f/f];*Vav1-Cre* mice but not in *Cic*[f/f];*Cd4-Cre* mice, which express significant amounts of CIC proteins in DP thymocytes. Moreover, CIC loss substantially suppressed TCR signaling in DP thymocytes, but not in SP cells. Although *Spry4*, *Dusp4*, *Dusp6*, and *Spred1* were all significantly derepressed in SP and DP cells from *Cic*[f/f];*Vav1-Cre* mice, among these four CIC target genes, *Spry4* and *Dusp6* were more dramatically derepressed

in DP cells than in SP cells in the absence of CIC (*Figure 7B*). These results suggest that *Spry4* and *Dusp6* may be CIC target genes primarily responsible for the CIC deficiency-mediated dysregulation of TCR signaling in DP cells and thymic selection processes. However, removal of the *Spry4* alleles only partially recovered the TCR signal intensity and positive selection, indicating that CIC regulates multiple target genes, including *Spry4*, to control TCR signaling and thymic T cell development. Notably, the hyperactivation of peripheral T cells and expansion of the Tfh cell population in *Cic*<sup>f/f</sup>;*Vav1-Cre* mice were also not rescued in *Spry4*<sup>-/-</sup>;*Cic*<sup>f/f</sup>;*Vav1-Cre* mice (*Figure 7—figure supplement 1C and D*). These findings suggest that the partial recovery of the TCR signal intensity and positive selection process by loss of SPRY4 was insufficient to ameliorate the abnormal peripheral T cell phenotypes in *Cic*<sup>f/f</sup>;*Vav1-Cre* mice or that derepression of CIC target genes other than *Spry4* could lead to T cell hyperactivation and enhanced Tfh cell formation in *Cic*<sup>f/f</sup>;*Vav1-Cre* mice. Concordantly, we have shown that CIC deficiency promotes Tfh cell differentiation via *Etv5* repression (*Park et al., 2017*). In contrast, CIC deficiency-induced thymic T cell phenotypes were not rescued in *Cic*<sup>f/f</sup>;*Etv5*<sup>f/f</sup>;*Vav1-Cre* mice (data not shown). Overall, these findings suggest that CIC regulates various target genes with differential effects to broadly control thymic T cell development and peripheral T cell activation and differentiation.

This study provides insight into how CIC controls autoimmunity. Because defects in thymic selection lead to the breakdown of central tolerance (*Xing and Hogquist, 2012*), it is conceivable that CIC deficiency during thymic T cell development generates autoreactive T cells, inducing autoimmunity. Therefore, CIC potentially suppresses autoimmunity by controlling the thymic selection process, as well as Tfh cell differentiation (*Park et al., 2017*). The more severe autoimmune-like phenotypes in *Cic*<sup>f/f</sup>;*Vav1-Cre* mice than in *Cic*<sup>f/f</sup>;*Cd4-Cre* mice (*Park et al., 2017*) highlight the significant contribution of the impaired thymic selection process to CIC deficiency-induced autoimmunity. To clarify the importance of each regulatory step in the suppression of autoimmunity, it will be necessary to better understand the molecular mechanisms underlying CIC regulation of thymic T cell development and Tfh cell differentiation. Additionally, studies on the role of CIC in T cell development and Tfh cell differentiation in humans should be conducted in the future to improve our understanding of the pathogenesis of autoimmune diseases such as systemic lupus erythematosus.

# Materials and methods

## Key resources table

| Reagent type (species) or resource | Designation | Source or reference | Identifiers | Additional information |
|---|---|---|---|---|
| Strain, strain background (*Mus musculus*) | C57BL/6 J | The Jackson Laboratory | RRID:IMSR_JAX:000664 | |
| Strain, strain background (*M. musculus*) | B6.Cg-Commd10<sup>Tg(Vav1-icre)A2Kio</sup>/J | The Jackson Laboratory | RRID:IMSR_JAX:008610 | |
| Strain, strain background (*M. musculus*) | B6.Cg-Tg(Cd4-cre)1Cwi/BfluJ | The Jackson Laboratory | RRID:IMSR_JAX:022071 | |
| Strain, strain background (*M. musculus*) | B6NTac.Cg-Tg(Lck-cre)1Cwi/Mmnc | *Lee et al., 2001* | RRID:MMRRC_037396-UNC | |
| Strain, strain background (*M. musculus*) | Cic<sup>floxed</sup> | *Lu et al., 2017*; *Park et al., 2017* | RRID:IMSR_JAX:030555 | |
| Strain, strain background (*M. musculus*) | Cic<sup>FLAG/FLAG</sup> | *Park et al., 2019* | PMID:30810242 | |
| Strain, strain background (*M. musculus*) | B6.Cg-Tg(TcraH-Y,TcrbH-Y)71Vbo | *Kisielow et al., 1988* | RRID:MGI:3588781 | |
| Strain, strain background (*M. musculus*) | B6.Cg-Tg(TcraTcrb)425Cbn/J | The Jackson Laboratory | RRID:IMSR_JAX:004194 | |
| Strain, strain background (*M. musculus*) | B6.Cg-Foxp3tm2Tch/J | The Jackson Laboratory | RRID:IMSR_JAX:006772 | |
| Strain, strain background (*M. musculus*) | Spry4<sup>-/-</sup> | This paper | N/A | generated using Spry4<sup>tm1a(KOMP)Mbp</sup> embryonic stem cells obtained from the UC Davis KOMP repository. |

*Continued on next page*

*Continued*

| Reagent type (species) or resource | Designation | Source or reference | Identifiers | Additional information |
|---|---|---|---|---|
| Cell line (*Homo-sapiens*) | Platinum-E (Plat-E) Retroviral Packaging Cell Line | Cell Biolabs | Cat# RV-101, RRID:CVCL_B488 | |
| Antibody | Anti-mouse CD3ε (Armenian Hamster monoclonal) | BioLegend Tonbo Biosciences | Cat# 100329, RRID:AB_1877171 Cat# 50–0031, RRID:AB_2621730 | FC (1:300) |
| Antibody | Anti-mouse CD4 (Rat monoclonal) | Biolegend BD Biosciences | Cat# 100552, RRID:AB_2563053 Cat# 562891, RRID:AB_2737870 | FC (1:300) |
| Antibody | Anti-mouse CD5 (Rat monoclonal) | Biolegend eBioscience | Cat# 100625, RRID:AB_2563928 Cat# 45-0051-80, RRID:AB_914332 | FC (1:300) |
| Antibody | Anti-mouse CD8α (Rat monoclonal) | BioLegend | Cat# 100723, RRID:AB_389304 Cat# 100721, RRID:AB_312760 | FC (1:300) |
| Antibody | Anti-mouse CD11b (Rat monoclonal) | BioLegend | Cat# 101211, RRID:AB_312794 | FC (1:300) |
| Antibody | Anti-mouse CD11c (Armenian Hamster monoclonal) | BioLegend | Cat# 117309, RRID:AB_313778 | FC (1:300) |
| Antibody | Anti-mouse CD19 (Rat monoclonal) | BD Biosciences | Cat# 561738, RRID:AB_10893995 | FC (1:300) |
| Antibody | Anti-mouse CD24 (Rat monoclonal) | BioLegend | Cat# 101819, RRID:AB_572010 | FC (1:300) |
| Antibody | Anti-mouse CD25 (Rat monoclonal) | Tonbo Biosciences | Cat# 75–0251, RRID:AB_2621943 | FC (1:300) |
| Antibody | Anti-mouse CD44 (Rat monoclonal) | BD Biosciences | Cat# 553133, RRID:AB_2076224 Cat# 561860, RRID:AB_10895375 | FC (1:300) |
| Antibody | Anti-mouse CD62L (Rat monoclonal) | BD Biosciences | Cat# 560516, RRID:AB_1645257 | FC (1:300) |
| Antibody | Anti-mouse CD69 (Armenian Hamster monoclonal) | BioLegend | Cat# 104513, RRID:AB_492844 Cat# 104545, RRID:AB_2686969 | FC (1:300) |
| Antibody | Anti-mouse CD73 (Rat monoclonal) | Biolegend eBioscience | Cat# 127223, RRID:AB_2716102 Cat# 12-0731-81, RRID:AB_763516 | FC (1:300) |
| Antibody | Anti-mouse CD90.1 (Mouse monoclonal) | eBioscience | Cat# 45-0900-82, RRID:AB_2573662 | FC (1:300) |
| Antibody | Anti-mouse CD90.2 (Rat monoclonal) | BioLegend | Cat# 140303, RRID:AB_10642686 | FC (1:300) |
| Antibody | Anti-mouse CXCR5 (Rat monoclonal) | BD Biosciences | Cat# 551960, RRID:AB_394301 | FC (1:100) |
| Antibody | Anti-mouse TCRβ (Armenian Hamster monoclonal) | Tonbo Biosciences | Cat# 35–5961, RRID:AB_2621723 | FC (1:300) |
| Antibody | Anti-mouse TCR Vβ2 (Rat monoclonal) | BioLegend | Cat# 127908, RRID:AB_1227784 | FC (1:300) |
| Antibody | Anti-mouse TCR Vβ5.1/5.2 (Mouse monoclonal) | BD Biosciences | Cat# 553189, RRID:AB_394697 | FC (1:300) |
| Antibody | Anti-mouse TCR Vβ6 (Rat monoclonal) | BioLegend | Cat# 140004, RRID:AB_10643583 | FC (1:300) |
| Antibody | Anti-mouse TCR Vβ7 (Rat monoclonal) | BioLegend | Cat# 118308, RRID:AB_893628 | FC (1:300) |
| Antibody | Anti-mouse TCR Vβ8.1/8.2 (Rat monoclonal) | BioLegend | Cat# 118408, RRID:AB_1134109 | FC (1:300) |
| Antibody | Anti-mouse TCR Vβ8.3 (Armenian Hamster monoclonal) | BD Biosciences | Cat# 553664, RRID:AB_394980 | FC (1:300) |
| Antibody | Anti-mouse TCR Vβ9 (Mouse monoclonal) | BioLegend | Cat# 139804, RRID:AB_10641563 | FC (1:300) |
| Antibody | Anti-mouse TCR Vβ11 (Rat monoclonal) | BioLegend | Cat# 125907, RRID:AB_1227781 | FC (1:300) |
| Antibody | Anti-mouse TCR Vβ12 (Mouse monoclonal) | BioLegend | Cat# 139704, RRID:AB_10639729 | FC (1:300) |
| Antibody | Anti-mouse TCR Vβ13 (Mouse monoclonal) | BioLegend | Cat# 140704, RRID:AB_10639945 | FC (1:300) |
| Antibody | Anti-mouse TCRγ/δ (Armenian Hamster monoclonal) | eBioscience | Cat# 17-5711-81, RRID:AB_842757 | FC (1:300) |

*Continued on next page*

*Continued*

| Reagent type (species) or resource | Designation | Source or reference | Identifiers | Additional information |
|---|---|---|---|---|
| Antibody | Anti-mouse NK-1.1 (Mouse monoclonal) | BioLegend | Cat# 108709, RRID:AB_313396 | FC (1:300) |
| Antibody | Anti-mouse TER-119 (Rat monoclonal) | eBioscience | Cat# 17-5921-81, RRID:AB_469472 | FC (1:300) |
| Antibody | Anti-mouse Gr-1 (Rat monoclonal) | eBioscience | Cat# 17-5931-81, RRID:AB_469475 | FC (1:300) |
| Antibody | Anti-mouse TCR H-Y (Mouse monoclonal) | eBioscience | Cat# 11-9930-81, RRID:AB_465452 | FC (1:300) |
| Antibody | Anti-mouse PD-1 (Rat monoclonal) | eBioscience | Cat# 11-9981-81, RRID:AB_465466 | FC (1:300) |
| Antibody | Anti-mouse Nur77 (Mouse monoclonal) | eBioscience | Cat# 12-5965-82, RRID:AB_1257209 | FC (1:100) |
| Antibody | Anti-T-bet (Mouse monoclonal) | eBioscience | Cat# 12-5825-80, RRID:AB_925762 | FC (1:100) |
| Antibody | Anti-RORγt (Rat monoclonal) | eBioscience | Cat# 12-5825-80, RRID:AB_925762 | FC (1:100) |
| Antibody | Anti-Foxp3 (Rat monoclonal) | eBioscience | Cat# 17-5773-80, RRID:AB_469456 | FC (1:100) |
| Antibody | Anti-DYKDDDDK(flag) Tag antibody (Rat monoclonal) | BioLegend | Cat# 637309, RRID:AB_2563147 | FC (1:100) |
| Antibody | PE Donkey anti-rabbit IgG (min. x-reactivity) antibody (Donkey Polyclonal) | BioLegend | Cat# 406421, RRID:AB_2563484 | FC (1:100) |
| Antibody | Anti-CIC (Rabbit polyclonal) | *Kim et al., 2015* | PMID:25653040 | WB (1:1000) |
| Antibody | PLCγ1 (D9H10) XP Rabbit mAb antibody (Rabbit monoclonal) | Cell Signaling Technology | Cat# 5690, RRID:AB_10691383 | WB (1:1000) |
| Antibody | Phospho-PLC 1 (Tyr783) antibody (Rabbit polyclonal) | Cell Signaling Technology | Cat# 2821, RRID:AB_330855 | WB (1:500) |
| Antibody | Zap-70 (99F2) Rabbit mAb antibody (Rabbit monoclonal) | Cell Signaling Technology | Cat# 2705, RRID:AB_2273231 | WB (1:1000) |
| Antibody | Phospho-Zap-70 (Tyr319)/Syk (Tyr352) antibody (Rabbit polyclonal) | Cell Signaling Technology | Cat# 2701, RRID:AB_331600 | WB (1:500) |
| Antibody | SAPK/JNK antibody (Rabbit polyclonal) | Cell Signaling Technology | Cat# 9252, RRID:AB_2250373 | WB (1:2000) |
| Antibody | Phospho-SAPK/JNK (Thr183/Tyr185) antibody (Rabbit polyclonal) | Cell Signaling Technology | Cat# 9251, RRID:AB_331659 | WB (1:1000) |
| Antibody | p44/42 MAPK (Erk1/2) antibody (Rabbit polyclonal) | Cell Signaling Technology | Cat# 9102, RRID:AB_330744 | WB (1:2000) |
| Antibody | Phospho-p44/42 MAPK (Erk1/2) (Thr202/Tyr204) antibody (Rabbit monoclonal) | Cell Signaling Technology | Cat# 4370, RRID:AB_2315112 | WB (1:1000) FC (1:100) |
| Antibody | Anti-p38 MAPK antibody (Rabbit polyclonal) | Cell Signaling Technology | Cat# 9212, RRID:AB_330713 | WB (1:2000) |
| Antibody | Phospho-p38 MAPK (Thr180/ Tyr182) antibody (Rabbit polyclonal) | Cell Signaling Technology | Cat# 9211, RRID:AB_331641 | WB (1:1000) |
| Antibody | β-Actin Antibody (Mouse monoclonal) | Santa Cruz Biotechnology | Cat# sc-47778, RRID:AB_626632 | WB (1:1000) |
| Antibody | α-Tubulin antibody (A-6) (Mouse monoclonal) | Santa Cruz Biotechnology | Cat# sc-398103, RRID:AB_2832217 | WB (1:1000) |
| Antibody | Purified NA/LE Hamster Anti-Mouse CD3e (Armenian Hamster monoclonal) | BD Biosciences | Cat# 553057, RRID:AB_394590 | (10 μg/mL) |
| Antibody | AffiniPure Goat Anti-Armenian Hamster IgG (H + L) (Goat polyclonal) | Jackson ImmunoResearch Labs | Cat# 127-005-160, RRID:AB_2338972 | (25 μg/mL) |
| Recombinant DNA reagent | pMIGR1-GFP (plasmid) | *Pear et al., 1998* | RRID:Addgene_27490 | |

*Continued on next page*

*Continued*

| Reagent type (species) or resource | Designation | Source or reference | Identifiers | Additional information |
|---|---|---|---|---|
| Recombinant DNA reagent | pMIGR1-SPRY4-GFP (plasmid) | This paper | N/A | CDS of mouse *Spry4* was inserted into pMIGR1-GFP. |
| Recombinant DNA reagent | pMIGR1-DUSP6-GFP (plasmid) | This paper | N/A | CDS of mouse *Dusp6* was inserted into pMIGR1-GFP. |
| Recombinant DNA reagent | pCL-Eco (plasmid) | *Naviaux et al., 1996* | RRID:Addgene_12371 | Retrovirus packaging vector |
| Sequence-based reagent | primers used in qRT-PCR | This paper | | See *Supplementary file 3* for sequence information. |
| Peptide, recombinant protein | Streptavidin | Southern Biotech | Cat# 7100–01 | |
| Peptide, recombinant protein | Streptavidin | eBioscience BD Biosciences | Cat# 45-4317-80, RRID:AB_10260035 Cat# 554067, RRID:AB_10050396 | FC (1:100) |
| Commercial assay or kit | Fixable Viability Dye eFluor 780 | eBioscience | Cat# 65-0865-14 | |
| Commercial assay or kit | Ghost Dye Violet 510 | Tonbo Biosciences | Cat# 13–0870 | |
| Commercial assay or kit | Foxp3/ Transcription Factor Staining Buffer Set | eBioscience | Cat# 00-5523-00 | |
| Commercial assay or kit | BD Cytofix Fixation Buffer | BD Biosciences | Cat# 554655, RRID:AB_2869005 | |
| Commercial assay or kit | FuGENE HD Transfection Reagent | Promega | Cat# E2311 | |
| Commercial assay or kit | RiboEx | GeneAll | Cat# 301–002 | |
| Commercial assay or kit | GoScript Reverse Transcriptase Kit | Promega | Cat# A5001 | |
| Commercial assay or kit | SYBR Green Realtime PCR Master Mix | TOYOBO | Cat# TOQPK-201 | |
| Commercial assay or kit | EasySep Mouse Streptavidin RapidSpheres Isolation Kit | Stem Cell Technologies | Cat# 19,860 | |
| Commercial assay or kit | BCA Protein Assay Kit | Pierce | Cat# 23,225 | |
| Commercial assay or kit | Clarity Western ECL Substrate | Bio-Rad | Cat# 1705061 | |
| Commercial assay or kit | SuperSignal West Dura Extended Duration Substrate | Thermo Scientific | Cat# 34,076 | |
| Chemical compound, drug | Indo-1, AM, cell permeant | Invitrogen | Cat# I1203 | |
| Chemical compound, drug | Ionomycin from Streptomyces conglobatus | Sigma-Aldrich | Cat# I9657-1MG | |
| Chemical compound, drug | Hexadimethrine bromide | Sigma-Aldrich | Cat# H9268-10G | |
| Software, algorithm | FlowJo | Tree Star Inc. | RRID:SCR_008520, https://www.flowjo.com/solutions/flowjo | |
| Software, algorithm | ImageJ | NIH | RRID:SCR_003070, https://imagej.nih.gov/ij/ | |
| Software, algorithm | GraphPad Prism 7 | GraphPad Software | RRID:SCR_002798, https://www.graphpad.com/scientific-software/prism/ | |

## Mice

All mice were maintained on a C57BL/6 background. *Cic*-floxed (*Lu et al., 2017*; *Park et al., 2017*), *Vav1-Cre* (*de Boer et al., 2003*), *Cd4-Cre* (*Lee et al., 2001*), p*Lck-Cre* (*Lee et al., 2001*), FLAG-tagged *Cic* knock-in (*Cic*$^{FLAG/FLAG}$) (*Park et al., 2019*), H-Y TCR transgenic (*Kisielow et al., 1988*), and OT-II TCR transgenic (*Barnden et al., 1998*) mice have been described previously. *Spry4*$^{-/-}$ mice were generated using *Spry4*$^{tm1a(KOMP)Mbp}$ embryonic stem cells obtained from the UC Davis KOMP repository. All experiments were conducted with age-matched mice, and 7–9 week-old mice were used unless otherwise indicated. Mice of both sexes were randomly allocated to the experimental groups. All mice were maintained in a specific pathogen-free animal facility under a standard 12 hr light/12 hr

dark cycle. Mice were fed standard rodent chow and provided with water ad libitum. All experiments were approved by the Institutional Animal Care and Use Committee of Pohang University of Science and Technology.

## Cell line

The Platinum-E (Plat-E) retroviral packaging cell line (Cell Biolabs) was grown in Dulbecco's modified Eagle's medium (DMEM, Welgene) supplemented with 10% fetal bovine serum (FBS, Welgene) and penicillin/streptomycin (Gibco). The cells were cultured in a 37 °C incubator with 5% $CO_2$. Mycoplasma contamination was routinely tested using the e-Myco plus Mycoplasma Detection Kit (INtRON Bio).

## Flow cytometry

Surface staining was performed using the following fluorescence-labeled antibodies against: CD3 (145–2 C11; Tonbo Biosciences), CD4 (RM4-5; Biolegend, BD Biosciences), CD5 (53–7.3; Biolegend, eBioscience), CD8α (53–6.7; Biolegend), CD11b (M1/70; Biolegend), CD11c (N418, Biolegend), CD19 (1D3; BD Biosciences), CD24 (M1/69; Biolegend), CD25 (PC61.5; Tonbo Biosciences), CD44 (IM7, BD Biosciences), CD62L (MEL-14; BD Biosciences), CD69 (H1.2F3; Biolegend), CD73 (TY/11.8; eBioscience), CD90.1 (HIS51; eBioscience), CD90.2 (53–2.1; Biolegend), CXCR5 (2G8; BD Biosciences), TCRβ (H57-597; Tonbo Biosciences), TCR Vβ2 (B20.6; Biolegend), TCR Vβ5.1/5.2 (MR9-4; BD Biosciences), TCR Vβ6 (RR4-7; Biolegend), TCR Vβ7 (TR310; Biolegend), TCR Vβ8.1/8.2 (KJ16-133.18; Biolegend), TCR Vβ8.3 (1B3.3; BD Biosciences), TCR Vβ9 (MR10-2; Biolegend), TCR Vβ11 (KT11; Biolegend), TCR Vβ12 (MR11-1; Biolegend), TCR Vβ13 (MR12-4; Biolegend), TCRγ/δ (GL-3; eBioscience), NK-1.1 (PK136; Biolegend), TER-119 (TER-119; eBioscience), Gr-1 (RB6-8C5; eBioscience), TCR H-Y (T3.70; eBioscience), and PD-1 (RMP1-30; eBioscience). For CXCR5 staining, cells were incubated with biotinylated anti-CXCR5 for 30 min and then sequentially incubated with APC- or PerCP- Cy5.5-labelled streptavidin (eBioscience) with other surface antibodies. Live/dead staining was performed using Fixable Viability Dye (FVD) eFluor 780 (eBioscience) or Ghost Dye Violet 510 (Tonbo Biosciences). Intracellular staining was performed using the Foxp3 staining buffer set (eBioscience). For intracellular staining, fluorochrome-labeled antibodies to FLAG (L5; Biolegend), T-bet (4B10; eBioscience), RORγt (B2D; eBioscience), Foxp3 (FJK-16s; eBioscience), and Nur77 (12.14; eBioscience) were used. For phospho-ERK staining, cells were stained with FVD for 30 min on ice, fixed with BD Cytofix for 30 min at room temperature, and permeabilized with cold methanol for at least 30 min at –20 °C. Permeabilized cells were stained with the p-ERK antibody (Cell Signaling Technology), and fluorochrome-labeled antibodies to surface markers and secondary antibody against rabbit IgG (Biolegend) were added. The stained cells were analyzed using an LSRII Fortessa flow cytometer (BD Biosciences) or CytoFLEX LX (Beckman Coulter). Data were analyzed using FlowJo software (TreeStar).

## Cell sorting

Single-cell suspensions of thymocytes were stained for surface markers including lineage (TCRγ/δ, NK-1.1, TER-119, Gr-1, CD11b, CD11c, CD19), lin/DN (CD4-CD8-), ISP (CD4-CD8+TCRβloCD24hi), DP (CD4+CD8+TCRβlo), CD4+ SP (CD4+CD8-TCRβhi), and CD8+ SP (CD4-CD8+TCRβhi) cells were sorted. To sort DN3 (lin-CD4-CD8-CD44loCD25hi) and DN4 (lin-CD4-CD8-CD44loCD25lo) cells, CD8- thymocytes were obtained by negative selection using the EasySep Mouse Streptavidin Rapid Spheres Isolation Kit (Stem Cell Technologies) and cell sorting was performed. A MoFlo-XDP (Beckman Coulter) was used for cell sorting.

## Generation of BM chimeric mice

To create a mixed BM chimera, $1 \times 10^6$ BM cells from each donor were mixed and injected intravenously into C57BL/6 recipient mice that had been irradiated (10 Gy). After 8 weeks of recovery, the mice were sacrificed, and thymi were harvested and homogenized to prepare single-cell suspensions. The cells were then stained using flow cytometry.

## Analysis of Nur77 expression

To measure Nur77 expression, freshly isolated thymocytes at $1 \times 10^7$ cells/ml were incubated with plate-coated anti-CD3 (5 μg/ml) and anti-CD28 (10 μg/ml) for 2 hr, followed by staining of surface

markers (CD4, CD8, and TCRβ) and intracellular staining of Nur77. Samples were analyzed using an LSRII Fortessa flow cytometer or CytoFLEX LX. Data were analyzed using FlowJo software.

## In vitro TCR stimulation

Total thymocytes or sorted DP cells were rested for 30 min in Roswell Park Memorial Institute (RPMI) medium (Welgene) at 37 °C. Cells were then washed with T-cell medium (TCM, RPMI supplemented with 10% FBS [Welgene], 1% penicillin/streptomycin [Gibco], and 0.1% β-mercaptoethanol [Gibco]) and incubated with biotin-conjugated anti-CD3 (60 µg/ml, 145–2 C11) and anti-CD4 (60 µg/ml, GK1.5) for 20 min on ice. Cells were washed and incubated for 5 min on ice with streptavidin (60 µg/ml, SouthernBiotech) for cross-linking and incubated for the indicated time at 37 °C. To analyze p-ERK levels in virus-transduced thymocytes by flow cytometry, unconjugated anti-CD3 (10 µg/ml, 145–2 C11) and goat anti-hamster IgG (25 µg/ml, Jackson Immunoresearch) were used to stimulate TCR. The cells were incubated for 2 min at 37 °C for TCR stimulation. One milliliter of Cold PBS was added at the end of stimulation and cell pellets were lysed for western blotting or further stained with anti-p-ERK antibody for flow cytometry.

## Calcium influx measurement

Freshly isolated thymocytes or virus-transduced thymocytes were incubated with 4 µM Indo-1-AM (Invitrogen) in TCM at 37 °C for 40 min. Cells were washed twice with TCM and incubated with soluble anti-CD3 (10 µg/ml) and fluorochrome-conjugated antibodies for surface markers (CD4, CD8, and TCRβ) in TCM on ice for 20 min. The cells were then washed with TCM and warmed before cross-linking. Goat anti-hamster IgG (25 µg/ml) was added to cross-link anti-CD3, and the signals were measured by flow cytometry. Ionomycin was added to ensure that the T cells were effectively loaded with Indo-1. The emission wavelength ratios of $Ca^{2+}$-bound to unbound Indo-1 were analyzed using an LSRII Fortessa flow cytometer.

## Plasmids and retroviral transduction

The coding sequences (CDSs) of mouse *Spry4* or *Dusp6* with enzyme sites XhoI/EcoRI were amplified by PCR and cloned into the MigR1 retroviral vector (*Pear et al., 1998*). Viruses were generated through transient cotransfection of Plat-E cells with cloned retroviral vectors and the pCL-Eco helper plasmid (Imgenex) (*Naviaux et al., 1996*). Briefly, $2.5 \times 10^6$ Plat-E cells were plated in 100 mm plates. The next day, cells were transfected with 3.6 µg of retroviral vector and 2.4 µg of pCL-Eco using FuGENE HD transfection reagent (Promega). Retrovirus-containing supernatants were harvested twice at 48 h and 72 h after transfection and filtered with a 0.22 µm syringe filter (Millipore). For retroviral transduction, $10^7$ thymocytes from wild-type adult mice were mixed with 0.5 ml of the viral supernatant and 1.5 ml of fresh TCM in the presence of 4 µg/ml polybrene (Sigma) and seeded in a well of a six-well plate for spin-infection at 1000 *g* for 90 min at 25 °C. After spin infection, the cells were incubated for 48 hr before harvesting.

## Western blotting

Sorted cells or stimulated cells were lysed in RIPA buffer (50 mM Tris-HCl pH 7.4, 150 mM NaCl, 1 mM PMSF, 1% NP-40, 0.5% sodium deoxycholate, 0.1% SDS, 1 x Roche Complete Protease Inhibitor Cocktail, and 1 x Roche Phosphatase Inhibitor Cocktail). Protein concentrations were measured using a BCA kit (Pierce). Equal amounts of protein samples were separated by 9% SDS-PAGE and transferred onto nitrocellulose membranes (Bio-Rad). The following primary antibodies were used: anti-CIC (homemade) (*Kim et al., 2015*), anti-PLCγ1 (#5690, Cell Signaling), anti-p-PLCγ1 (#2821, Cell Signaling), anti-ZAP-70 (#2705, Cell Signaling), anti-p-ZAP-70 (#2701, Cell Signaling Technology), anti-JNK (#9252, Cell Signaling), anti-p-JNK (#9251, Cell Signaling), anti-ERK (#9102, Cell Signaling), anti-p-ERK (#4370, Cell Signaling), anti-p38 (#9212, Cell Signaling), anti-p-p38 (#9211, Cell Signaling), anti-β-actin (#sc-47778, Santa Cruz), and anti-α-tubulin (#sc-398103, Santa Cruz). Membranes were incubated with secondary antibodies conjugated to horseradish peroxidase (HRP) and developed using Clarity Western ECL Substrate (Bio-Rad) or SuperSignal West Dura (Thermo Scientific). Images were acquired using an ImageQuant LAS 500 instrument (GE Healthcare).

## RNA isolation, cDNA synthesis, and qRT-PCR

Total RNA was extracted from sorted cells using RiboEx (GeneAll) and 0.2–1 μg was subjected to cDNA synthesized using the GoScript Reverse Transcription system (Promega) according to the manufacturer's instructions. SYBR Green real-time PCR master mix (TOYOBO) was used for qRT-PCR analysis. The primers used for qRT-PCR are listed in *Supplementary file 3*.

## RNA sequencing and data analysis

Thymi from $Cic^{f/f}$ and $Cic^{f/f}$;*Vav1-Cre* mice were dissected and homogenized into single-cell suspensions for cell sorting. DP thymocytes were sorted on the basis of surface markers of lin⁻CD4⁺CD8⁺T-CRβ$^{lo}$, and total RNA was extracted with RiboEx. The library for mRNA sequencing was generated using the TruSeq Stranded Total RNA LT Sample Prep Kit (Illumina), and sequencing was performed using NovaSeq 6,000 (Illumina). Trimmed reads were mapped to the mouse reference genome (mm10 RefSeq) using HISAT2, and the transcripts were assembled using StringTie. DEGs were generated using edgeR and genes with fold changes > 2 and *P*-values < 0.05, were selected for Gene Ontology (GO) analyses on the basis of biological processes using the DAVID website (https://david.ncifcrf.gov/).

## Statistical Analysis

Statistical analyses were performed using GraphPad Prism 7 (https://www.Graphpad.com, La Jolla, CA, USA). All experiments were independently performed more than three times. Two-tailed Student's *t-tests* were used to obtain P-values between the two groups. One-way or two-way ANOVA with Tukey's multiple comparison test was used to calculate P-values among the three different groups. Statistical significance was set at $P < 0.05$. Error bars indicate standard error of the mean (SEM).

## Acknowledgements

We thank Dr. Jaeho Cho and the Lee lab members for their helpful discussions and comments on this study. This work was supported by *grants from the Samsung Science and Technology Foundation* under project number SSTF-BA1502-14 and the National Research Foundation (NRF) of Korea (NRF-2021R1A2C3004006 and –2017 R1A5A1015366). JSP and JP were supported by BK21. HH was supported by a Global PhD Fellowship (NRF-2017H1A2A1042705).

## Additional information

### Funding

| Funder | Grant reference number | Author |
| --- | --- | --- |
| Samsung Science and Technology Foundation | SSTF-BA1502-14 | Yoontae Lee |
| National Research Foundation of Korea | NRF-2021R1A2C3004006 | Yoontae Lee |
| National Research Foundation of Korea | NRF-2017R1A5A1015366 | Yoontae Lee |
| National Research Foundation of Korea | NRF-2017H1A2A1042705 | Hyebeen Hong |
| Brain Korea 21 | | Jong Seok Park Jiho Park |

The funders had no role in study design, data collection and interpretation, or the decision to submit the work for publication.

### Author contributions

Soeun Kim, Conceptualization, Data curation, Formal analysis, Investigation, Methodology, Project administration, Validation, Visualization, Writing - original draft, Writing - review and editing; Guk-Yeol Park, Jong Seok Park, Jiho Park, Hyebeen Hong, Investigation; Yoontae Lee, Conceptualization,

Funding acquisition, Project administration, Supervision, Writing - original draft, Writing - review and editing

**Author ORCIDs**
Soeun Kim (iD) http://orcid.org/0000-0002-6425-0899
Yoontae Lee (iD) http://orcid.org/0000-0002-6810-3087

**Ethics**
Animal experimentation: All experiments were approved by the Institutional Animal Care and Use Committee of Pohang University of Science and Technology (POSTECH-2019-0081). All experiments were carried out in accordance with the approved guidelines. Mouse sacrifice was performed under isoflurane anesthesia, and every effort was made to minimize suffering.

**Decision letter and Author response**
Decision letter https://doi.org/10.7554/eLife.71769.sa1
Author response https://doi.org/10.7554/eLife.71769.sa2

---

## Additional files

**Supplementary files**
• Supplementary file 1. List of Differentially expressed genes (DEGs) in *Cic*-deficient DP thymocytes.
• Supplementary file 2. Gene Ontology (GO) analysis of DEGs in *Cic*-deficient DP thymocytes.
• Supplementary file 3. Oligonucleotide sequences used for qRT-PCR.
• Supplementary file 4. Flow cytometry gating strategy.
• Transparent reporting form

**Data availability**
The Gene Expression Omnibus (GEO) accession number for the RNA sequencing data of DP thymocytes reported in this paper is GSE173909. All data generated or analysed during this study are included in the manuscript and supporting files. Source data files have been provided for Figures 1, 2, 3, 4, 5, 6, and 7.

The following dataset was generated:

| Author(s) | Year | Dataset title | Dataset URL | Database and Identifier |
|---|---|---|---|---|
| Lee Y, Kim S | 2021 | RNA Sequencing Analysis of Gene Expression Profiles in WT and CIC-deficient DP Thymocytes | https://www.ncbi.nlm.nih.gov/geo/query/acc.cgi?acc=GSE173909 | NCBI Gene Expression Omnibus, GSE173909 |

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
