## [Editor Report]

This paper focuses on the transcriptional regulation of the T cell receptor (TCR) signaling cascade and would be of interest to those studying T cell development and differentiation. The authors employ a conditional deletion of the Capicua (Cic) gene, a transcriptional repressor previously shown to be involved in regulating autoimmunity and follicular helper T (Tfh) cell differentiation, and now show that loss of CIC in hematopoietic cells leads to defects in TCR-β selection as well as in positive and negative selection of developing thymocytes. The overall conclusions are well supported by the findings.

---

## [Decision Letter]

**Decision letter after peer review:**

Thank you for submitting your article "Regulation of positive and negative selection and TCR signaling during thymic T cell development by capicua" for consideration by *eLife*. Your article has been reviewed by 3 peer reviewers, including JC Zúñiga-Pflücker as the Reviewing Editor and Reviewer #1, and the evaluation has been overseen by Tadatsugu Taniguchi as the Senior Editor.

Essential revisions:

1) The main shared concerns were about how the data is presented and interpreted; additional information/discussion is required to fully appreciate the quality and appropriateness of the conclusions. The discussion disproportionately focuses on potential defects in beta-selection and thymic Treg differentiation that were only superficially studied in this manuscript. At this stage, perhaps the interpretation of these results should be tempered.

2) Please address the recommendations to the authors listed below by the reviewers, with an emphasis on Ref #3 concerns, as well as the corrected highlighted by Ref #2.

*Reviewer #1 (Recommendations for the authors):*

1. The main conclusion is that TCR signaling strength is affected by the increased expression of antagonists of the ERK/MAPK signaling pathway, as well as calcium, which are clearly supported by the observations shown. One would also predict that γ/δ TCR signaling to be affected during development, and the authors should show whether gd T cell development or differentiation is also affected, in particular as it pertains to the preTCR vs gd-TCR checkpoint.

2. One surprising finding is that in the bone marrow chimera experiments, when using equal numbers of WT and Cic-deficient cells, that the thymocytes seem to have equal contributions of WT and Cic-deficient cells, as shown in Figure 6A. However, the actual frequencies of Wt vs Cic-/- cells are not shown and should be included. If these are similar, it would be curious given the defect in preTCR signaling , which typically accounts for a large number of downstream DP thymocytes. This should be further discussed in the text.

*Reviewer #2 (Recommendations for the authors):*

– Figure 1C: FACS plots are not representative of the graphs (seems like CD4-cre mice display an intermediate phenotype).

– Figure 1D has no legend.

– Figure 1D : FACS plot doesn't reflect the graphs (increased DP cell population in CD4-cre mice).

– Figure 1E: These results suggest that a portion of the SP thymocytes in older mice may be recirculating SP cells. The authors should do an analysis of SP thymocyte maturation (show TCRβ vs CD24 for example) to investigate this. Also, they should determine if the DN3/4 block is seen in young mice.

– Figure 2: Is CIC protein absent in DN1/2 thymocytes of Vav-cre mice?

– Figure 5 : It seems odd that Calcium flux is affected by CIC deletion and not PLCγ phosphorylation. Do the authors have a theory why?

– Figure 6 : To avoid transferring other immune cells like B and DC and answer the question the author are asking with this experiment, bone marrow cells must be lineage depleted before injection. Was this the case? (method section mention transfer of whole BM).

– pLCK model is interesting to discriminate impact of CIC deletion on DN cells from DP cells. Authors should describe this model and show if they recapitulate the DP/SP phenotype without the DN stages being affected.

– Since the authors base their explanation of the auto-immune phenotype on impaired thymocyte selection in the Vav-cre model, why is there no impact on thymocyte development but still signs of autoimmunity in CD4-cre mice, as mentioned in the previous paper? Authors should discuss these discrepancies.

*Reviewer #3 (Recommendations for the authors):*

1) Spry4 (and DUSP6) as a target responsible for much of the phenotype observed in CIC-deficient thymocytes is intriguing. However, it is not clear to this reviewer how the over-expression studies were performed. What is the efficiency of retroviral transduction in, presumably adult, thymocytes? Given nearly full suppression of p-ERK and calcium induction after activation in cells reported to be over-expressing Spry4 (or DUSP6), presumably the transduced cells were tracked – what marker was used?

2) There is a rather striking decrease in CD5 on DP cells on TCR transgenic CIC-deficient DP cells. Is this due solely to differences in the proportion of DP cells undergoing thymic selection (e.g. CD69+) or because of differences in the strength of signal itself? Indeed, differences in CD5 on polyclonal DP cells from Cicf/f;Vav1-Cre versus control DP are also quite striking. While the (presumably) post-selection DP cells expressing higher levels of CD5 seem to express similar levels of this molecule, the bulk population (presumably pre-selection) is significantly lower. What do the authors make of this? Are the majority of DP cells sensing no TCR stimulation at all? It seems as if this might be the case even with ex vivo stimulation; at least based on the provided histogram, a larger percentage of DP thymocytes appear to fail to upregulate Nur77 after in vitro stimulation while those cells that do upregulate Nur77 seem to do so similar levels whether CIC-deficient or control. Further, why is TCR signaling more impacted at the DP stage than at the SP stage in the absence of CIC?

3) There appear to be differences in the extent to which CIC impacts CD4^+^ and CD8^+^ SP thymocyte numbers at different stages of ontogeny. The authors imply that mature T cells that have recirculated to the thymus in adult mice (the 7 week old mice presented in this manuscript or as previously reported in 9 week old mice) may mask any striking differences in the relative proportions and numbers of CD4^+^ and CD8^+^ SP thymocytes in CIC-deficient as compared to control mice. In younger mice, there are significantly fewer mature thymocytes in the absence of CIC. Whether this is due to differences in the recirculated mature T cell population is less clear than implied; this could be due to differences in the selection processes that accompany T cell development at different stages of ontogeny. One would need to use appropriate markers (e.g. CD73) or reporters (Rag-GFP) to make this distinction.

4) Careful explanation of the experimental set up and conclusions from the TCR sequencing studies would be appreciated. I do not understand the argument for the longer CDR3 sequences in the CIC KO conventional CD4 T cell populations as being 'pre-selection-like'; what does this imply? It appears as if the main conclusion of the TCR sequencing data is that the differences in the repertoire predominantly lie in the Treg population; outside of TCR sequence analysis this subset is not analyzed in the current manuscript. Are there overt differences in thymic Treg development in the absence of CIC?

5) Gating strategies and representative flow plots, as well as clear descriptions of the gates in the figure legends, for all analyses would be appreciated. It is not always clear, for example, if lineage+ cells have been removed from DN gates, whether mature T cells have been gated on TCRbhi cells, whether the conventional CD4^+^ SP population used for TCR sequencing includes CD25+ Treg progenitors, etc. In addition, representative histograms are not always provided for MFI analysis; this is important to understand, for example with the CIC-Flag tag, the extent to which expression is heterogenous in a population; clear statements about the population for which MFI is calculated (e.g. for Figure 4B, is the MFI calculated for the population in the positive gate or for the total population) should be added.

6) Consider splitting some of the data onto separate graphs (e.g. Figure 3C and D) as it is very difficult to appreciate the noted significant differences in terms of percentages and cell numbers when the symbols are against the x axis, for example.

7) Please revisit the appropriateness of the t test for assessing statistical significance across three mouse strains.

8) Please ensure that biological and experimental replicates are clearly noted for each experiment. For example, how many mice were used for the TCR sequencing experiments?

[Editors' note: further revisions were suggested prior to acceptance, as described below.]

Thank you for resubmitting your work entitled "Regulation of positive and negative selection and TCR signaling during thymic T cell development by capicua" for further consideration by *eLife*. Your revised article has been reviewed by 3 peer reviewers, one of whom is a member of our Board of Reviewing Editors, and the evaluation has been overseen by Tadatsugu Taniguchi as the Senior Editor.

The manuscript has been improved but there are some remaining issues that need to be addressed, as outlined below:

The remaining required revisions are clearly outlined within the detailed reviewers comments below.

*Reviewer #1 (Recommendations for the authors):*

The authors have comprehensively addressed most of my initial concerns, however additional points need to be clarified. In particular, the differences in pLck-cre vs Vav-cre mice should be better addressed more clearly.

*Reviewer #2 (Recommendations for the authors):*

The authors have provided satisfactory responses to points #1, 2, 3, 4, 5 and 7 but this reviewer still has a problem with the answer to comment #6: The authors claim that , as they expected, no defect was observed in DP/SP frequencies of CIC f/f pLCK-cre mice. However, in contrast to the CD4-cre model, CIC depletion seems complete in DP thymocytes of CIC f/f pLCK-cre mice (Figure S6). Do the authors have an explanation for why they don't observe the same DP/SP phenotype (defects) in the Vav-cre and pLCK-cre models?

*Reviewer #3 (Recommendations for the authors):*

I appreciate the author responses to previous questions and critiques; the manuscript is improved though some outstanding issues remain.

1. The integration of some of the new data is unconventional. Additional analysis (including supplemental figures) of Treg development in Cic cKO mice as well as mature T cell recirculation to the thymus appears to be added to the discussion rather than the Results section. Following this, the explanation for differences in the phenotypes of 1, 7, and 9 week-old mice based on the absence of a recirculated T cell phenotype in WT vs Cic cKO mice at 9 weeks is not clear to me.

2. The authors now make it clear that the TCR sequencing datasets are n=1. While their data interpretation is consistent with their hypothesis, I am concerned about making conclusions on this sample set.

3. Additional information is provided for the thymocyte transduction protocol and subsequent analysis; yet, ambiguities remain. It appears as if the thymocytes (from adult mice) were transduced in the absence of incubation with cytokines, and the transduction rate seems rather high for this population as described. Perhaps more details are needed. In addition, though the authors show a representative example of the GFP in a Supplementary file, given a BD Cytofix followed by cold methanol protocol is reported prior to p-ERK staining, I wonder about the extent to which GFP is preserved for this staining condition (these reagents have been reported quench fluorescence under some conditions and for at least some GFP variants).

4. Some gating strategies were clarified while others are still ambiguous to this reviewer. For example, in some cases CD8 SP analyses include pre-gating on TCRb+ cells. This does not seem to be the case in all figures, however. For example, for the quantification of CD8 SP cells in 4C, I wonder if these are ISPs and the interpretation of the results is skewed.

---

## [Author Response]

Essential revisions:1) The main shared concerns were about how the data is presented and interpreted; additional information/discussion is required to fully appreciate the quality and appropriateness of the conclusions. The discussion disproportionately focuses on potential defects in beta-selection and thymic Treg differentiation that were only superficially studied in this manuscript. At this stage, perhaps the interpretation of these results should be tempered.

We appreciate this kind and helpful suggestion. We have shortened our discussion on the topic of beta-selection and thymic Treg cells, and tempered the interpretation of the results.

2) Please address the recommendations to the authors listed below by the reviewers, with an emphasis on Ref#3 concerns, as well as the corrected highlighted by Ref #2.

We have addressed every suggestion raised by the three reviewers by conducting additional experiments and data analyses. We hope that our point-by-point responses to the reviewers’ comments are clear and satisfactory.

Reviewer #1 (Recommendations for the authors):1. The main conclusion is that TCR signaling strength is affected by the increased expression of antagonists of the ERK/MAPK signaling pathway, as well as calcium, which are clearly supported by the observations shown. One would also predict that γ/δ TCR signaling to be affected during development, and the authors should show whether gd T cell development or differentiation is also affected, in particular as it pertains to the preTCR vs gd-TCR checkpoint.

As suggested, we analyzed thymic γδT cells in 7-week-old *Cic^f/f^;Vav1-Cre* mice. We found a slight increase in the frequency of total γδT cells in the thymus of *Cic^f/f^;Vav1-Cre* mice. However, the frequencies of mature thymic CD24^lo^CD44^hi^ γδT cells and T-bet-expressing type 1 and RORγt-expressing type 17 γδT cells were comparable between WT and *Cic^f/f^;Vav1-Cre* mice. These results are presented in Figure 1—figure supplement 1 and are described in the Results section (page 6, lines 122-126).

2. One surprising finding is that in the bone marrow chimera experiments, when using equal numbers of WT and Cic-deficient cells, that the thymocytes seem to have equal contributions of WT and Cic-deficient cells, as shown in Figure 6A. However, the actual frequencies of Wt vs Cic-/- cells are not shown and should be included. If these are similar, it would be curious given the defect in preTCR signaling , which typically accounts for a large number of downstream DP thymocytes. This should be further discussed in the text.

We thank the reviewer for pointing out this important issue and apologize for the confusion caused. We analyzed the frequencies of Thy1.1/Thy1.2 WT and Thy1.1/Thy1.1 CIC-deficient thymocytes in mixed bone marrow chimeric mice. The results are presented next to the representative FACS plot in Figure 6A. We found that the frequency of CIC-deficient cells was significantly lower than that of WT cells, which may support the idea of defective pre-TCR signaling in CIC-deficient thymocytes.

Reviewer #2 (Recommendations for the authors):– Figure 1C: FACS plots are not representative of the graphs (seems like CD4-cre mice display an intermediate phenotype).– Figure 1D has no legend.– Figure 1D : FACS plot doesn't reflect the graphs (increased DP cell population in CD4-cre mice).

We thank the reviewer for these thoughtful comments. We replaced the FACS plots in Figure 1C and D with more representative images. A legend for Figure 1D has been added at the correct position.

– Figure 1E: These results suggest that a portion of the SP thymocytes in older mice may be recirculating SP cells. The authors should do an analysis of SP thymocyte maturation (show TCRβ vs CD24 for example) to investigate this. Also, they should determine if the DN3/4 block is seen in young mice.

We appreciate this valuable suggestion. Reviewer #3 also asked a similar question about the issue of recirculating SP cells (Reviewer #3’s comment no. 3). We examined the frequency of CD24^lo^CD73^+^ recirculating CD4^+^ SP cells in the thymus of WT and *Cic^f/f^;Vav1-Cre* mice at 9 weeks of age. The proportion of this cell population was comparable between WT and *Cic^f/f^;Vav1-Cre* mice, suggesting that the similar frequency of SP thymocytes in WT and *Cic^f/f^;Vav1-Cre* mice at 9 weeks of age did not result from an accumulation of higher numbers of recirculated peripheral SP cells in the thymus of *Cic^f/f^;Vav1-Cre* mice. The corresponding data are presented in Figure 1—figure supplement 3. We also included the interpretation of these data in the Discussion section (pages 17 and 18, lines 385-404). To address the second question, we examined the frequency of DN subsets in the thymus of 1-week-old mice. In agreement with results from 7-week-old mice, a partial block of the DN3-to-DN4 transition was observed in 1-week-old *Cic^f/f^;Vav1-Cre* mice. The results are presented in Figure 1—figure supplement 2B.

– Figure 2: Is CIC protein absent in DN1/2 thymocytes of Vav-cre mice?

Accordingly, we examined CIC protein levels in DN1/2 cells by western blotting and confirmed the absence of CIC in DN1/2 thymocytes of *Cic^f/f^;Vav1-Cre* mice. This result has been added to Figure 2A.

– Figure 5 : It seems odd that Calcium flux is affected by CIC deletion and not PLCγ phosphorylation. Do the authors have a theory why?

We thank the reviewer for this valuable question. It has been reported that SPRY4 suppresses VEGF-A-induced PIP_2_ breakdown and calcium flux without affecting PLCγ activation (Ayada et al., 2009). We have mentioned this in the corresponding Results section of the revised manuscript (page 14, lines 325-326).

– Figure 6 : To avoid transferring other immune cells like B and DC and answer the question the author are asking with this experiment, bone marrow cells must be lineage depleted before injection. Was this the case? (method section mention transfer of whole BM).

As mentioned in the Methods section, we used whole BM from WT and *Cic^f/f^;Vav1-Cre* mice (50:50) to generate mixed BM chimeras. We agree that transferring lineage-depleted BM cells is a standard way to generate BM chimeras. However, we believe that it is still valid to examine T cell-intrinsic functions of CIC in mice transferred with a 50:50 mixture of whole BM from WT and *Cic^f/f^;Vav1-Cre* mice, because WT and CIC-deficient hematopoietic stem and progenitor cells develop into T cell subsets within the same thymic environment. Numerous studies have successfully determined the cell-intrinsic function of a specific gene product by examining mixed BM chimeric mice transferred with whole BM cells (Huang et al., 2021; Redd et al., 2018; Ulges et al., 2015).

– pLCK model is interesting to discriminate impact of CIC deletion on DN cells from DP cells. Authors should describe this model and show if they recapitulate the DP/SP phenotype without the DN stages being affected.

Accordingly, we analyzed the frequency of DN, DP, and SP cells in the thymus of WT and *Cic^f/f^;*p*Lck-Cre* mice at 7 weeks of age, and presented the results in Figure 6—figure supplement 1D and E. As expected, thymic T cell development was normal during the DN stage in *Cic^f/f^;*p*Lck-Cre* mice. However, a decrease in the frequency of SP thymocytes was not unambiguously observed in 7-week-old *Cic^f/f^;*p*Lck-Cre* mice. This result is consistent with our conclusion that *Cic^f/f^;*p*Lck-Cre* mice have a milder defect in TCR signaling and positive selection than *Cic^f/f^;Vav1-Cre* mice (Figure 6—figure supplement 1A and B).

– Since the authors base their explanation of the auto-immune phenotype on impaired thymocyte selection in the Vav-cre model, why is there no impact on thymocyte development but still signs of autoimmunity in CD4-cre mice, as mentioned in the previous paper? Authors should discuss these discrepancies.

We appreciate this important question. In our previous study (Park et al., 2017), we showed that both *Cic^f/f^;Vav1-Cre* and *Cic^f/f^;Cd4-Cre* mice had a systemic autoimmune-like phenotype with expansion of the follicular helper T (Tfh) cell population and that derepression of *Etv5* promoted Tfh cell differentiation in CIC-deficient mice. Since excessive formation and/or hyperactivation of Tfh cells is closely associated with the onset of autoimmunity (Crotty, 2019), systemic autoimmunity in *Cic^f/f^;Cd4-Cre* mice could be attributed to the *Etv5* derepression-mediated promotion of Tfh cell differentiation. We found that CIC deficiency-induced autoimmune-like symptoms were significantly reduced in *Cic^f/f^;Etv5^f/f^;Cd4-Cre* mice (unpublished data), validating our previous conclusion. Our findings of defective thymic T cell development in *Cic^f/f^;Vav1-Cre* mice explain why *Cic^f/f^;Vav1-Cre* mice have a more severe autoimmune-like phenotype than *Cic^f/f^;Cd4-Cre* mice with normal thymic T cell development. We have mentioned this in the Discussion section (page 20, lines 444-458).

Reviewer #3 (Recommendations for the authors):1) Spry4 (and DUSP6) as a target responsible for much of the phenotype observed in CIC-deficient thymocytes is intriguing. However, it is not clear to this reviewer how the over-expression studies were performed. What is the efficiency of retroviral transduction in, presumably adult, thymocytes? Given nearly full suppression of p-ERK and calcium induction after activation in cells reported to be over-expressing Spry4 (or DUSP6), presumably the transduced cells were tracked – what marker was used?

We apologize for the lack of clarity. We have thus eliminated any ambiguities regarding the experimental procedure performed for analyzing retrovirus-transduced thymocytes. We infected thymocytes from adult WT mice with retrovirus co-expressing GFP and either SPRY4 or DUSP6, and subsequently analyzed the levels of phospho-ERK and calcium flux in GFP-positive DP cells. The virus infection efficiency was approximately 20%. We clarified this point in the corresponding figure legend and Supplementary file 4I (gating strategy for GFP-positive DP thymocytes).

2) There is a rather striking decrease in CD5 on DP cells on TCR transgenic CIC-deficient DP cells. Is this due solely to differences in the proportion of DP cells undergoing thymic selection (e.g. CD69+) or because of differences in the strength of signal itself? Indeed, differences in CD5 on polyclonal DP cells from Cicf/f;Vav1-Cre versus control DP are also quite striking. While the (presumably) post-selection DP cells expressing higher levels of CD5 seem to express similar levels of this molecule, the bulk population (presumably pre-selection) is significantly lower. What do the authors make of this? Are the majority of DP cells sensing no TCR stimulation at all? It seems as if this might be the case even with ex vivo stimulation; at least based on the provided histogram, a larger percentage of DP thymocytes appear to fail to upregulate Nur77 after in vitro stimulation while those cells that do upregulate Nur77 seem to do so similar levels whether CIC-deficient or control. Further, why is TCR signaling more impacted at the DP stage than at the SP stage in the absence of CIC?

Questions about:

1) CD5 levels in WT and CIC-deficient CD69^-^ and CD69^+^ DP cells:

To directly address the reviewer’s question, we examined the frequency of CD69^-^ and CD69^+^ DP cells in WT and CIC-deficient mice. We also compared CD5 levels in CD69^-^ and CD69^+^ DP cells between WT and CIC-deficient mice. As expected, the frequency of CD69^+^ DP cells was lower in CIC-deficient mice than in WT mice (Figure 5B, Figure 5—figure supplement 1B and E). Importantly, a substantial decrease in CD5 levels was detected in CIC-deficient CD69^-^ DP cells, but not in CD69^+^ DP cells, compared to WT cells (Figure 5C, Figure 5—figure supplement 1C and F). As most DP cells were CD69^-^ (more than 95% of total DP cells; Figure 5B, Figure 5—figure supplement 1B and E), decreased CD5 levels in CIC-deficient DP cells probably resulted from attenuated TCR signaling in DP cells rather than a decrease in the frequency of CD69^+^ DP cells in CIC-deficient mice.

2) Histograms of Nur77 expression in DP thymocytes:

As the reviewer pointed out, Nur77 expression analysis in DP cells resulted in a two-peaked histogram (Figure 4B, Figure 7—figure supplement 1B). However, there was a difference between WT and CIC-deficient DP cells at the second peak representing an upregulation of Nur77 upon TCR stimulation (Figure 4B, Figure 7—figure supplement 1B). For Nur77 expression analysis, we treated thymocytes at 1×10^7^ cells/ml with plate-coated anti-CD3 (5 μg/ml) and anti-CD28 (10 μg/ml) antibodies for 2 h to stimulate the TCR, followed by intracellular staining of Nur77. By conducting additional experiments, we found that the appearance of the two peaks depended on the experimental conditions used. We performed the same experiments using various concentrations of thymocytes. The appearance of two peaks was reproduced when thymocytes were employed at 1×10^7^ cells/ml (Author response image 1). Interestingly, the first peak corresponding to the Nur77-negative DP cell population gradually disappeared as the concentration of thymocytes was reduced (Author response image 1). Therefore, we concluded that the histogram of Nur77 expression could be modified by the experimental conditions employed, such as concentrations of thymocytes, anti-CD3, and anti-CD28, as well as the incubation time taken. We would also like to mention that another research group using a method similar to ours obtained a similar histogram pattern for Nur77-expressing cells (Thien et al., 2010) (Author response image 1).

**Author response image 1. sa2fig1:** Histograms of Nur77 expression in double-positive (DP) thymocytes (A) Flow cytometric analysis of Nur77 expression in DP thymocytes. Various concentrations of freshly isolated thymocytes were incubated with plate-coated anti-CD3 (5 μg/ml) and anti-CD28 (10 μg/ml) antibodies for 2 h, followed by staining of surface markers (CD4, CD8, and TCRβ) and intracellular staining of Nur77. Representative histograms are presented. (B) Independent experiment showing two-peaked histograms of Nur77-expressing cells. Adapted from Thien et al. (2010).

3) DP cell-specific regulation of TCR signaling by CIC:

We do agree that this is an important question. We believe that the qRT-PCR data presented in Figure 7B may partially explain this question. CIC deficiency more dramatically derepressed *Spry4* and *Dusp6*, CIC target genes responsible for inhibition of TCR signaling, in DP cells than in SP cells (Figure 7B), potentially affecting the TCR signaling pathway in DP cells more drastically than in SP cells. Moreover, our data showed that TCR signaling was weakly activated in DP cells compared to SP cells, when treated with the same concentration of anti-CD3 antibody (Figure 5A and D). In this regard, it can be inferred that the CIC deficiency-mediated derepression of CIC target genes may have had a significant inhibitory effect on TCR signaling in DP cells with weak activation of TCR signaling, but a negligible inhibitory effect on TCR signaling in SP cells with strongly activated TCR signaling. We have included this in the Discussion section (pages 18 and 19, lines 405-431).

3) There appear to be differences in the extent to which CIC impacts CD4^+^ and CD8^+^ SP thymocyte numbers at different stages of ontogeny. The authors imply that mature T cells that have recirculated to the thymus in adult mice (the 7 week old mice presented in this manuscript or as previously reported in 9 week old mice) may mask any striking differences in the relative proportions and numbers of CD4^+^ and CD8^+^ SP thymocytes in CIC-deficient as compared to control mice. In younger mice, there are significantly fewer mature thymocytes in the absence of CIC. Whether this is due to differences in the recirculated mature T cell population is less clear than implied; this could be due to differences in the selection processes that accompany T cell development at different stages of ontogeny. One would need to use appropriate markers (e.g. CD73) or reporters (Rag-GFP) to make this distinction.

We thank the reviewer for this valuable suggestion. Accordingly, we analyzed the frequency of recirculated mature CD4^+^ T cells in the thymus of WT and *Cic^f/f^;Vav1-Cre* mice at 9 weeks of age using CD73 as a recirculating T cell marker. Unexpectedly, the frequency of thymic CD24^lo^CD73^+^CD4^+^ SP cells was comparable between WT and *Cic^f/f^;Vav1-Cre* mice (Figure 1—figure supplement 3). Thereafter, we seriously considered the reviewer’s comment that the decreased thymic SP cell population in 1-week-old *Cic^f/f^;Vav1-Cre* mice could have resulted from differences in the selection processes that accompany T cell development at different stages of ontogeny. Based on a detailed research of the literature, we realized that negative selection is inefficient early in ontogeny and increases with age (He et al., 2013; Huseby et al., 2001). Since *Cic^f/f^;Vav1-Cre* mice have defects in both positive and negative selection, it is conceivable that the decreased frequency of SP thymocytes in 1-week-old *Cic^f/f^;Vav1-Cre* mice was caused by a defect in positive selection, and that this effect was attenuated by ineffective negative selection at 7 weeks of age or older. We discuss this interesting and reasonable possibility in the Discussion section (pages 17 and 18, lines 385-404).

4) Careful explanation of the experimental set up and conclusions from the TCR sequencing studies would be appreciated. I do not understand the argument for the longer CDR3 sequences in the CIC KO conventional CD4 T cell populations as being 'pre-selection-like'; what does this imply? It appears as if the main conclusion of the TCR sequencing data is that the differences in the repertoire predominantly lie in the Treg population; outside of TCR sequence analysis this subset is not analyzed in the current manuscript. Are there overt differences in thymic Treg development in the absence of CIC?

Previous studies have demonstrated that T cells with shorter CDR3 sequences are enriched in mature post-selection T cell populations (Hou et al., 2019; Lu et al., 2019). Based on this knowledge, we suggest that the increased frequency of longer CDR3 sequences in the TCR repertoires of CIC-deficient CD4^+^ SP thymocytes might have been the result of defective selection processes. As suggested, we analyzed thymic Treg cell populations in 7-week-old WT and *Cic^f/f^;Vav1-Cre* mice. Consistent with a defect in negative selection in *Cic^f/f^;Vav1-Cre* mice, the frequency of thymic CD25^+^Foxp3^+^ Treg and CD25^-^Foxp3^lo^ progenitor cells was significantly increased in *Cic^f/f^;Vav1-Cre* mice. The corresponding data are presented in Figure 4—figure supplement 2 and are mentioned in the Discussion section (page 19, lines 440-441).

5) Gating strategies and representative flow plots, as well as clear descriptions of the gates in the figure legends, for all analyses would be appreciated. It is not always clear, for example, if lineage+ cells have been removed from DN gates, whether mature T cells have been gated on TCRbhi cells, whether the conventional CD4^+^ SP population used for TCR sequencing includes CD25+ Treg progenitors, etc. In addition, representative histograms are not always provided for MFI analysis; this is important to understand, for example with the CIC-Flag tag, the extent to which expression is heterogenous in a population; clear statements about the population for which MFI is calculated (e.g. for Figure 4B, is the MFI calculated for the population in the positive gate or for the total population) should be added.

We appreciate this thoughtful suggestion. Accordingly, we provided gating strategies for all flow cytometry experiments in Supplementary file 4 and described the gates in the figure legends. We also show representative flow plots and histograms for all flow cytometry and MFI analyses, including histograms for CIC-FLAG expression in various thymic T cell subsets (Figure 1A). To answer some of the reviewer’s specific questions, we removed lineage^+^ cells for all DN cell analyses, and excluded CD25^+^ Treg progenitors from the conventional CD4^+^ SP population for TCR repertoire analysis of non-Treg cells. We presented the graphs for Nur77-associated MFI for both total and Nur77^+^ DP cells in Figure 4B and Figure 7—figure supplement 1B.

6) Consider splitting some of the data onto separate graphs (e.g. Figure 3C and D) as it is very difficult to appreciate the noted significant differences in terms of percentages and cell numbers when the symbols are against the x axis, for example.

We thank the reviewer for this thoughtful suggestion. As suggested, the frequency and number of DN, DP, 4SP, and 8SP cells in Figure 3C, 3D, and 4C are displayed in separate graphs for reasons of clarity and comprehensibility.

7) Please revisit the appropriateness of the t test for assessing statistical significance across three mouse strains.

Accordingly, we used one-way or two-way ANOVA with Tukey’s multiple comparison test for reanalyzing the statistical significance across three groups of mice or samples.

8) Please ensure that biological and experimental replicates are clearly noted for each experiment. For example, how many mice were used for the TCR sequencing experiments?

As per the reviewer’s suggestion, we clearly indicated biological and experimental replicates for each experiment in the corresponding figure legends and/or method sections.

References:

Ayada, T., Taniguchi, K., Okamoto, F., Kato, R., Komune, S., Takaesu, G., and Yoshimura, A. (2009). Sprouty4 negatively regulates protein kinase C activation by inhibiting phosphatidylinositol 4,5-biphosphate hydrolysis. Oncogene 28, 1076–1088.

Crotty, S. (2019). T Follicular Helper Cell Biology: A Decade of Discovery and Diseases. Immunity 50, 1132–1148.

He, Q., Morillon, Y.M., Spidale, N.A., Kroger, C.J., Liu, B., Sartor, R.B., Wang, B., and Tisch, R. (2013). Thymic Development of Autoreactive T Cells in NOD Mice Is Regulated in an Age-Dependent Manner. J. Immunol. 191, 5858–5866.

Hou, X., Zeng, P., Zhang, X., Chen, J., Liang, Y., Yang, J., Yang, Y., Liu, X., and Diao, H. (2019). Shorter TCR β-chainsare highly enriched during thymic selection and antigen driven slection. Front. Immunol. 10, 299.

Huang, H., Zhou, P., Wei, J., Long, L., Shi, H., Dhungana, Y., Chapman, N.M., Fu, G., Saravia, J., Raynor, J.L., et al. (2021). in vivo CRISPR screening reveals nutrient signaling processes underpinning CD8^+^ T cell fate decisions. Cell 184, 1245-1261.e21.

Huseby, E.S., Sather, B., Huseby, P.G., and Goverman, J. (2001). Age-dependent T cell tolerance and autoimmunity to myelin basic protein. Immunity 14, 471–481.

Lu, J., Van Laethem, F., Bhattacharya, A., Craveiro, M., Saba, I., Chu, J., Love, N.C., Tikhonova, A., Radaev, S., Sun, X., et al. (2019). Molecular constraints on CDR3 for thymic selection of MHC-restricted TCRs from a random pre-selection repertoire. Nat. Commun. 10, 1–14.

Park, S., Lee, S., Lee, C.G., Park, G.Y., Hong, H., Lee, J.S., Kim, Y.M., Lee, S.B., Hwang, D., Choi, Y.S., et al. (2017). Capicua deficiency induces autoimmunity and promotes follicular helper T cell differentiation via derepression of ETV5. Nat. Commun. 8, 1–13.

Redd, P.S., Lu, C., Klement, J.D., Ibrahim, M.L., Zhou, G., Kumai, T., Celis, E., and Liu, K. (2018). H3K4me3 mediates the NF-κB p50 homodimer binding to the pdcd1 promoter to activate PD-1 transcription in T cells. Oncoimmunology 7, 1–11.

Thien, C.B.F., Dagger, S.A., Steer, J.H., Koentgen, F., Jansen, E.S., Scott, C.L., and Langdon, W.Y. (2010). c-Cbl promotes T cell receptor-induced thymocyte apoptosis by activating the phosphatidylinositol 3-kinase/Akt pathway. J. Biol. Chem. 285, 10969–10981.

Ulges, A., Klein, M., Reuter, S., Gerlitzki, B., Hoffmann, M., Grebe, N., Staudt, V., Stergiou, N., Bohn, T., Brühl, T.J., et al. (2015). Protein kinase CK2 enables regulatory T cells to suppress excessive TH2 responses in vivo. Nat. Immunol. 16, 267–275.

[Editors' note: further revisions were suggested prior to acceptance, as described below.]

The manuscript has been improved but there are some remaining issues that need to be addressed, as outlined below:The remaining required revisions are clearly outlined within the detailed reviewers comments below.Reviewer #1 (Recommendations for the authors):The authors have comprehensively addressed most of my initial concerns, however additional points need to be clarified. In particular, the differences in pLck-cre vs Vav-cre mice should be better addressed more clearly.

Thank you for this thoughtful suggestion. As per your suggestion, we carefully examined the phenotypes of *Cic^f/f^;*p*Lck-Cre* mice compared to those of WT, *Cic^f/f^;Cd4-Cre*, and *Cic^f/f^;Vav1-Cre* mice, and quantified CIC levels in developing thymic T cell subsets from WT, *Cic^f/f^;Cd4-Cre*, *Cic^f/f^;Vav1-Cre*, and *Cic^f/f^;*p*Lck-Cre* mice using ImageJ software (Figure 6—figure supplement 1A). It was apparent that *Cic^f/f^;*p*Lck-Cre* mice exhibited normal DN cell development (Figure 6—figure supplement 1B), but there were defects in positive selection and TCR signaling in DP cells (Figure 6—figure supplement 1C-E). However, these defects were milder in *Cic^f/f^;*p*Lck-Cre* mice than in *Cic^f/f^;Vav1-Cre* mice (Figure 6—figure supplement 1C-E), which could be attributed to the incomplete removal of CIC expression in DP thymocytes of *Cic^f/f^;*p*Lck-Cre* mice (Figure 6—figure supplement 1A). To more clearly explain the phenotypes of *Cic^f/f^;*p*Lck-Cre* mice in comparison with those of *Cic^f/f^;Vav1-Cre* mice, data were reorganized and the text in the corresponding Results section was amended accordingly (pages 13 and 14, lines 298-320).

Reviewer #2 (Recommendations for the authors):The authors have provided satisfactory responses to points #1, 2, 3, 4, 5 and 7 but this reviewer still has a problem with the answer to comment #6: The authors claim that , as they expected, no defect was observed in DP/SP frequencies of CIC f/f pLCK-cre mice. However, in contrast to the CD4-cre model, CIC depletion seems complete in DP thymocytes of CIC f/f pLCK-cre mice (Figure S6). Do the authors have an explanation for why they don't observe the same DP/SP phenotype (defects) in the Vav-cre and pLCK-cre models?

Thank you for raising this issue and apologies for how we carried out data analysis and interpretation. As described in the response to Reviewer #1’s comment, we quantified CIC levels in developing thymic T cell subsets from WT, *Cic^f/f^;Cd4-Cre*, *Cic^f/f^;Vav1-Cre*, and *Cic^f/f^;*p*Lck-Cre* mice (Figure 6—figure supplement 1A). Approximately 24% of CIC proteins were still expressed in DP cells from *Cic^f/f^;*p*Lck-Cre* mice compared to WT cells (Figure 6—figure supplement 1A), indicating incomplete depletion of CIC in DP cells of *Cic^f/f^;*p*Lck-Cre* mice. Similar to the findings in 7-week-old *Cic^f/f^;Vav1-Cre* mice (Figure 1D), a slight increase in the frequency of DN and DP thymocytes, and a decrease in the frequency of CD4^+^ SP cells was observed in 7 week-old *Cic^f/f^;*p*Lck-Cre* mice (Figure 6—figure supplement 1C). It is worth noting that the DP/SP phenotypes were also mild in *Cic^f/f^;Vav1-Cre* mice at 7 weeks old when compared to WT mice (a significant but slight decrease in the frequency of CD4^+^ SP thymocytes only) (Figure 1D). Therefore, it would be hard to expect that *Cic^f/f^;*p*Lck-Cre* mice will show significant changes in the frequency of DP and SP cells at 7 weeks old compared to WT mice because CIC depletion is incomplete in DP thymocytes of *Cic^f/f^;*p*Lck-Cre* mice. We rearranged the figure panels in Figure 6—figure supplement 1 and accordingly edited the text in the corresponding Results section (pages 13 and 14, lines 298-320) to make our conclusion clearer.

Reviewer #3 (Recommendations for the authors):I appreciate the author responses to previous questions and critiques; the manuscript is improved though some outstanding issues remain.1. The integration of some of the new data is unconventional. Additional analysis (including supplemental figures) of Treg development in Cic cKO mice as well as mature T cell recirculation to the thymus appears to be added to the discussion rather than the Results section. Following this, the explanation for differences in the phenotypes of 1, 7, and 9 week-old mice based on the absence of a recirculated T cell phenotype in WT vs Cic cKO mice at 9 weeks is not clear to me.

We appreciate this comment. Accordingly, we moved the data on the recirculating CD4^+^ SP cells in the thymus to the Results section (page 7, lines 142-153). We also analyzed the frequency of recirculating thymic CD4^+^ SP cells in 1-week-old WT and *Cic^f/f^;Vav1-Cre* mice and presented the data in Figure 1—figure supplement 3A. These modifications allowed us to more logically explain why we focused on the regulatory function of CIC in thymic T cell development.

The frequency and number of SP thymocytes were significantly decreased in *Cic^f/f^;Vav1-Cre* mice at 1 week old (Figure 1E), whereas these defects were attenuated and disappeared with age (Figure 1D) (Park et al., 2017). To determine whether the disappearance of the decrease in the frequency of SP thymocytes in *Cic^f/f^;Vav1-Cre* mice at 9 weeks old was due to increased accumulation of recirculating SP cells in the thymus of *Cic^f/f^;Vav1-Cre* mice compared to WT mice, we analyzed the frequency of recirculating CD24^lo^CD73^+^CD4^+^ SP cells in the thymus of 9-week-old WT and *Cic^f/f^;Vav1-Cre* mice. The comparable frequency of thymic recirculating CD4^+^ SP cells between 9-week-old WT and *Cic^f/f^;Vav1-Cre* mice (Figure 1—figure supplement 3B) suggests that the SP cell population recirculated into the thymus from the periphery was not the cause of the phenotypic changes in *Cic^f/f^;Vav1-Cre* mice with age.

2. The authors now make it clear that the TCR sequencing datasets are n=1. While their data interpretation is consistent with their hypothesis, I am concerned about making conclusions on this sample set.

We appreciate your concern regarding the reliability of data obtained from only one sample set. To validate the TCR repertoire analysis results, we performed flow cytometric analysis of thymic CD4^+^ non-Treg and Treg cells from WT and *Cic^f/f^;Vav1-Cre* mice using antibodies specific to various TCRβ V segments. To our surprise, the results were markedly different from our previous conclusions based on TCR repertoire sequencing analysis. Among the 10 different TCRβ V segments tested, the usage frequencies of eight TCRβ V segments in non-Treg cells were significantly different between WT and *Cic^f/f^;Vav1-Cre* mice, whereas only two TCRβ V segments were differentially used in the thymic Treg cell compartments of WT and *Cic^f/f^;Vav1-Cre* mice. The data are presented in Figure 4—figure supplement 2. Considering your concerns and these findings, we decided to exclude the TCR repertoire sequencing analysis data from the text. Accordingly, we revised the corresponding sections of the manuscript (page 11, lines 228-246). We believe that this decision does not affect the main conclusion of this study, and that the new data (Figure 4—figure supplement 2) still support our findings that CIC regulates thymic selection processes.

3. Additional information is provided for the thymocyte transduction protocol and subsequent analysis; yet, ambiguities remain. It appears as if the thymocytes (from adult mice) were transduced in the absence of incubation with cytokines, and the transduction rate seems rather high for this population as described. Perhaps more details are needed. In addition, though the authors show a representative example of the GFP in a Supplementary file, given a BD Cytofix followed by cold methanol protocol is reported prior to p-ERK staining, I wonder about the extent to which GFP is preserved for this staining condition (these reagents have been reported quench fluorescence under some conditions and for at least some GFP variants).

As suggested, we have expounded on the procedures of retroviral transduction of thymocytes and subsequent analysis in the Methods section of the revised manuscript (pages 22-23, lines 511-513; page 24, lines 554-555; page 25, lines 579-583). As pointed out, thymocytes from adult C57/BL6 mice were transduced with retroviral supernatant and incubated for 48 h in the absence of additional cytokines. Based on the frequency of GFP^+^ thymocytes in three independent experiments, the transduction efficiency varied in each trial (Author response image 2).

**Author response image 2. sa2fig2:** Efficiency of retroviral transduction of thymocytes. FACS plots showing the frequency of GFP^+^ thymocytes transduced with retrovirus co-expressing GFP and control, SPRY4, or DUSP6. Three independent experiments were performed.

We agree with the reviewer’s opinion that the GFP signal can be lost depending on the intracellular staining methods. We detected the loss of GFP signal when performing intracellular staining of transcription factors in thymocytes of *Foxp3^GFP^* mice using the Foxp3 staining buffer set from eBioscience (data not shown). However, we did not observe this phenomenon when performing intracellular staining of p-ERK in thymocytes transduced with GFP-expressing retrovirus using BD cytofix and cold methanol. To address the reviewer’s question, we investigated the extent of preserved GFP signal after permeabilization of thymocytes transduced with GFP-expressing retrovirus with the Foxp3 staining buffer set or cold methanol. Consistent with our previous observations, cell permeabilization with cold methanol did not significantly affect GFP signal in retrovirus-transduced thymocytes, whereas the Foxp3 staining buffer set (eBioscience) led to a dramatic decrease in the GFP signal (Author response image 3).

**Author response image 3. sa2fig3:** Comparison of the extent of preserved GFP signal after cell permeabilization by different methods. (A) Flow cytometric analysis of GFP expression in live thymocytes infected with GFP-expressing retrovirus. Representative FACS plots are presented. No perm: without permeabilization, eBioscience: the Foxp3 staining buffer set, and methanol: cold methanol. (B) Relative expression levels of GFP in thymocytes before and after permeabilization with the Foxp3 staining buffer set (eBioscience) or cold methanol. N=2 for each group.

4. Some gating strategies were clarified while others are still ambiguous to this reviewer. For example, in some cases CD8 SP analyses include pre-gating on TCRb+ cells. This does not seem to be the case in all figures, however. For example, for the quantification of CD8 SP cells in 4C, I wonder if these are ISPs and the interpretation of the results is skewed.

We appreciate this thoughtful comment. To address your concern, we analyzed the proportion of CD24^hi^TCRβ^lo^ ISP cells among total CD8^+^ SP thymocytes from female and male H-Y*;Cic^f/f^* and H-Y*;Cic^f/f^;Vav1-Cre* mice; and presented the data in Figure 3—figure supplement 1B and Figure 4—figure supplement 1B, respectively. CD24^hi^TCRβ^lo^ ISP cells constituted less than 10% of total CD8^+^ SP thymocytes in male H-Y mice, and their frequency was comparable between male H-Y*;Cic^f/f^* and H-Y*;Cic^f/f^;Vav1-Cre* mice (Figure 4—figure supplement 1B). Therefore, we concluded that the expansion of the thymic CD8^+^ SP cell population in male H-Y*;Cic^f/f^;Vav1-Cre* mice was primarily attributed to a defect in negative selection. In female mice, the frequency of ISP cells was significantly increased in H-Y*;Cic^f/f^;Vav1-Cre* mice compared to H-Y*;Cic^f/f^* mice (Figure 3—figure supplement 1B), demonstrating that the decreased frequency of CD8^+^ SP thymocytes was due to a defect in positive selection rather than attenuated ISP cell formation.

Reference:

Park, S., Lee, S., Lee, C.G., Park, G.Y., Hong, H., Lee, J.S., Kim, Y.M., Lee, S.B., Hwang, D., Choi, Y.S., et al. (2017). Capicua deficiency induces autoimmunity and promotes follicular helper T cell differentiation via derepression of ETV5. Nat. Commun. 8, 1–13.